*Prog Neurobiol.* 2021 December ; 207: 101887. doi:10.1016/j.pneurobio.2020.101887.

# Using high spatial resolution fMRI to understand representation in the auditory network

**Michelle Moerel**[a,b,c], **Essa Yacoub**[d], **Omer Faruk Gulban**[b,c,d,e], **Agustin Lage-Castellanos**[b,c,f], **Federico De Martino**[b,c,d,*]

[a]Maastricht Centre for Systems Biology, Maastricht University, Maastricht, the Netherlands

[b]Department of Cognitive Neuroscience, Faculty of Psychology and Neuroscience, Maastricht University, Maastricht, the Netherlands

[c]Maastricht Brain Imaging Center (MBIC), Maastricht, the Netherlands

[d]Center for Magnetic Resonance Research, Department of Radiology, University of Minnesota, Minneapolis, USA

[e]Brain Innovation B. V., Maastricht, the Netherlands

[f]Department of Neuroinformatics, Cuban Center for Neuroscience, Cuba

## Abstract

Following rapid methodological advances, ultra-high field (UHF) functional and anatomical magnetic resonance imaging (MRI) has been repeatedly and successfully used for the investigation of the human auditory system in recent years. Here, we review this work and argue that UHF MRI is uniquely suited to shed light on how sounds are represented throughout the network of auditory brain regions. That is, the provided gain in spatial resolution at UHF can be used to study the functional role of the small subcortical auditory processing stages and details of cortical processing. Further, by combining high spatial resolution with the versatility of MRI contrasts, UHF MRI has the potential to localize the primary auditory cortex in individual hemispheres. This is a prerequisite to study how sound representation in higher-level auditory cortex evolves from that in early (primary) auditory cortex. Finally, the access to independent signals across auditory cortical depths, as afforded by UHF, may reveal the computations that underlie the emergence of an abstract, categorical sound representation based on low-level acoustic feature processing. Efforts on these research topics are underway. Here we discuss promises as well as challenges that come with studying these research questions using UHF MRI, and provide a future outlook.

[*]Corresponding author at: Department of Cognitive Neuroscience, Faculty of Psychology and Neuroscience, Maastricht University, P.O. Box 616, 6200 MD Maastricht, the Netherlands. michelle.moerel@maastrichtuniversity.nl (M. Moerel).

Appendix A. The Peer Review Overview and Supplementary data

**Keywords**

Ultra-high field MRI; Auditory system; Subcortical processing; Laminar fMRI; Sound representation

## 1.    Introduction

Following rapid methodological advances, ultra-high field (UHF) magnetic resonance imaging (MRI) has emerged as a novel tool to study human audition. Here we argue that it can greatly aid the exploration of three research topics in human auditory neuroscience. First, the high spatial resolution offered by UHF MRI increases the accessibility to the subcortical auditory structures and their sub-nuclei, enabling the study of sound processing in these small brain regions. Importantly, high spatial resolution and large coverage together allow investigating subcortical and cortical sound representations (and their interaction/ changes across the pathway) within the same session, greatly facilitating the investigation of hierarchical brain processes. Second, the versatility of MR contrasts allows assessing a range of anatomical and functional brain characteristics and may thereby enable the in vivo and individually-based localization of human auditory cortical regions. This is not trivial, as the vast interindividual variation in superior temporal lobe macroanatomy severely limits the localization accuracy of auditory regions through probabilistic atlases. Third, UHF fMRI may shed light on column specific computations (i.e. across layers) that take place throughout the human auditory cortex. This may show how layer-dependent computations contribute to the emergence of an abstract, categorical sound representation from low-level acoustic feature processing. Together, insights into these topics are expected to fundamentally improve the mechanistic understanding of how sounds are represented in the network of auditory brain regions.

When moving from high (3 T; 3 T) to ultra-high magnetic fields at 7 T (7 T) and above, the blood-oxygenation-level-dependent (BOLD)-based susceptibility contrast, the basis of functional MRI (fMRI; Ogawa et al., 1992; Yacoub et al., 2001), increases. This results in higher signal-to-noise ratios (SNR; Vaughan et al., 2001) of the collected data, which can be leveraged for different purposes. Ultra-high field fMRI studies have, for example, used the SNR increase to shorten scan durations, potentially crucial for clinical applications. Alternatively, for the same scan duration that would be used at lower field strength, more stimuli could be presented at UHF. This facilitates the collection of a larger dataset in each individual, allowing, for example, for precision neuroscience (Gordon et al., 2017; Poldrack, 2017). Finally, the SNR gain can be used to acquire images with higher spatial resolution. While 3 T fMRI experiments are typically run with a spatial resolution of around 2–3 mm, at 7 T a spatial resolution below a millimeter can be achieved. Together with the increased specificity of the BOLD signal (Uludağ et al., 2009), the higher SNR enables the non-invasive exploration of previously inaccessible computational units in the human brain such as small subcortical structures and cortical depth-dependent processes (De Martino et al., 2016).

While the vast majority of 7 T fMRI studies to date have addressed research questions in visual neuroscience (Kok et al., 2016; Lawrence et al., 2018; Nasr et al., 2016; Olman et al., 2012; Polimeni et al., 2010; Yacoub et al., 2008), UHF MRI has been successfully applied to the study of the auditory system as well. In recent years, the application of UHF MRI to the auditory system allowed, for example, the examination of speech processing in the medial geniculate body (MGB; Mihai et al., 2019) and the exploration of columnar feature processing in auditory cortex (De Martino et al., 2015b; Moerel et al., 2018b). In 2017, the first 7 T MRI device was cleared by the U.S. Food and Drug Administration (FDA), and the EU approval followed suit. The first auditory clinical endeavors at ultra-high field strength are already available (Berlot et al., 2020; Ghazaleh et al., 2017). That is, tinnitus was investigated in two separate 7 T studies, where the increase in spatial resolution allowed mapping (changes in) frequency preference in the cortex of individual volunteers (Ghazaleh et al., 2017) and subcortically (in the IC and MGB; Berlot et al., 2020). Results suggested a reduced frequency selectivity as well as disturbed thalamocortical connectivity with tinnitus (Berlot et al., 2020). As technical challenges to facilitate the use of 7 T MRI in patient populations are being addressed (Trattnig et al., 2018), clinical applications of UHF MRI are expected to rapidly increase in feasibility and number in the years to come.

While clinical advances are promising and increasing, here we focus on how UHF MRI can aid the understanding of sound processing in healthy hearing individuals. Below we first review methodological challenges and then discuss the potential benefit of UHF MRI for investigating subcortical auditory processing, parcellating the human auditory cortex non-invasively, and studying the emergence of categorical responses in the auditory cortex.

## 2.   Auditory-specific methodological challenges at 7 T

Since the very first applications, the main challenge associated with the use of fMRI for auditory neuroscience research has been the loudness of the echo-planar-imaging (EPI) pulse sequence used for functional imaging (Talavage and Hall, 2012). Similarly to what happens in a loudspeaker where a wire (and a connected membrane) immersed in a magnetic field moves due to the Lorentz force when a current passes through it, so do the gradients in an MRI scanner move (expand and contract) when the currents are applied. These movements produce the loud noises heard when acquiring fMRI data using EPI. Scanners can produce noises that approach 120 dB SPL (Talavage and Hall, 2012) and the forces (and thus the noise) scales with the magnetic field strength (Schmitt et al., 1998). Resulting forces depend on the angle of the conductor relative to the magnetic field, as well as the current direction. The design of a gradient coil in which the forces were compensated locally (i.e., with multiple current generating loops that result in forces that cancel each other), was perhaps one of the first technological developments that benefitted auditory fMRI research. This gradient coil design reduced the overall vibration of the gradient coil assembly and thus the resulting acoustic noise (Mansfield et al., 1994). Initial efforts were also devoted to the design of systems that would allow an efficient delivery of the audio signals to the ear canal, while also attenuating the scanning noise (Ravicz and Melcher, 1998). Most systems used nowadays in both low and high field imaging achieve noise attenuation, compared to actual sound delivery, by combining the presentation of sounds in the ear canal with high quality earplugs and ear muffs (see e.g. the Sensimetric system; www.sens.com).

One of the most influential developments coming from the early auditory fMRI investigations was the introduction of the clustered volume acquisition technique (Edmister et al., 1999; Talavage et al., 1999; compare Fig. 1A-B), in which the acquisition of one fMRI volume (acquisition time – TA) is clustered within the time between the acquisition of two fMRI samples (volumes; repetition time – TR), leaving a silent gap for the presentation of sounds. While the BOLD response to the sound is still contaminated by the response to the scanner noise (i.e., the blue and red curves in Fig. 1B still overlap), the clustered volume acquisition technique has the advantage that sounds are clearly audible as they are presented in silence. This approach is also known as "sparse sampling" when combined with long TRs (Hall et al., 1999; Fig. 1C), and has the advantage of fully separating (in time) the BOLD response to the sound from the BOLD response to the scanner noise (the latter is not measured due to the very long TR). In a direct comparison between acquisition techniques, it was shown that the response amplitude to frequencies coinciding with scanner noise was underestimated by continuous and clustered acquisition, but not when using a sparse design (Langers et al., 2014). However, the use of very long TRs (i.e. very long gaps) results in the acquisition of only one brain volume measuring the response to a stimulus (presented in the gap), severely increasing the uncertainty of GLM-based BOLD estimates due to the reduced number of data points and thereby limiting the power of the experimental design. Moreover, researchers have to rely on assumptions regarding the BOLD time-to-peak at the moment of the study design (determining the placements of sounds in the gap relative to the subsequent acquisition; this can be partially overcome by jittering the placement of sound presentation within the TR). To overcome these limitations, previous studies have used a clustered volume acquisition technique with an intermediate inter-sound-interval duration (e.g. presenting a sound every few TRs). This introduces substantial variability in the measured response allowing separation of its sources, while maintaining a reasonable power (Fig. 1B). Alternatively, many volumes can be rapidly acquired after each silent gap (Fig. 1D). This technique, known as interleaved silent steady state (ISSS; Mueller et al., 2011; Schwarzbauer et al., 2006), is combined with a train of silent slice-selective excitation pulses during the gap (to avoid $T_1$-related signal decay) and has been used at 7 T for collecting cortical and subcortical responses (Riecke et al., 2018).

Some sequences lend themselves particularly well for clustered acquisitions. With the introduction of multi-band (MB) for EPI based fMRI, multiple 2D EPI slices can be collected simultaneously (Moeller et al., 2010; Setsompop et al., 2012), thereby permitting much shorter volume TRs. MB-EPI has become widely available and is routinely used. As this technique reduces the number of gradient 'noises' in the same TR, it can thus benefit clustered acquisition for auditory research (shortening the acquisition time and allowing for longer silent gaps) (De Martino et al., 2015a). It would be of interest to experimentally compare the sensitivity of continuous, clustered, and sparse acquisition techniques combined with fast temporal sampling as made feasible by MB EPI. Other MRI pulse sequence strategies, such as 3D approaches (i.e. 3D GRASE; Oshio and Feinberg, 1991), can acquire an entire volume during a single encoding period. Further, if reduced acquisition volumes or field of views are used, as is often the case for high resolution applications, the 3D acquisition time can be extremely short - in the order of a few hundred milliseconds. This in turn allows for the use of longer silent gaps for sound presentation (Box 1).

As an alternative to the noisy standard acquisition scheme, "silent" acquisition schemes have been proposed based on sinusoidal gradients (Peelle, 2014; Peelle et al., 2010; Schmitter et al., 2008; Zapp et al., 2012). Furthermore, readout schemes with continuous rather than pulsed noise have also been developed (Seifritz et al., 2006). However, the clustered volume acquisition (more recently applied together with MB excitation) has been used in the majority of the high field auditory fMRI studies conducted in recent years (Talavage and Hall, 2012). For example, while the selectivity to acoustic frequencies in auditory cortex had been revealed earlier at lower fields without the use of the clustered volume acquisition technique (Talavage et al., 2000), the clustered volume acquisition method with long silent gaps was at the basis of the first high field auditory fMRI study (i.e., sparse sampling; Formisano et al., 2003) that revealed mirror symmetric tonotopic maps. An exception to this is the required use of continuous acquisition when presenting long sound excerpts, such as audio books (Hanke et al., 2014) or movies (Uğurbil et al., 2013).

The auditory community continues to look with interest at further developments with a particular focus on ways to reduce the scanner noise. The use of active noise cancellation (Hall et al., 2009), for example, is intriguing. However, for an effective use at high fields it will have to be implemented in a system that is compatible with the coils that are routinely used (e.g. with limited space for large headphones). Apart from improved sound delivery systems, the development of alternative functional imaging techniques that do not use rapidly switching gradients can be of relevance. The minimal gradient switching steps used in sweep imaging with Fourier transform (SWIFT), for example (Idiyatullin et al., 2015, 2006), results in "quiet" functional imaging with no susceptibility induced dropouts typical of GE-EPI. SWIFT has the potential to become popular among auditory neuroscientists, who have been advocating for such features (Talavage and Hall, 2012).

When moving to ultra-high field, issues other than acquisition noise become more apparent. This especially holds true for imaging auditory cortical areas. In conventional volume transmission, the transmit ($B_1$) sensitivity decays more rapidly from the center to the outer regions in the imaged volume (i.e. the head) at UHF compared to lower field strength (Van de Moortele et al., 2009). As the auditory cortex is situated towards the outer region of the head, using volume coils require large increases in the transmit voltage in order to achieve the required flip angles. This issue is exacerbated when acquiring data with contrasts other than the conventional GE fMRI. At UHF, there are several options for collecting fMRI data (Table 1). The contribution of vasculature components to the acquired signal differs across these options. Gradient-echo echo-planar imaging (GE-EPI) is the most common technique for collecting fMRI data. The resulting signal is $T_2$*-weighted and contains contributions from both macro- and microvasculature, providing high sensitivity but a relatively lower specificity due to the biasing influence of large veins situated on the cortical pial surface and draining effects across cortical depths (Goense et al., 2007; Harel et al., 2006; Uludağ et al., 2009; Zhao et al., 2004). Instead, approaches dominated by a $T_2$-weighted signal (e.g., spin echo [SE] EPI, or 3D gradient echo and spin echo [3D GRASE; Oshio and Feinberg, 1991]) suppress the contribution of large vessels (Yacoub et al., 2003) and the signal originating from small veins is proportionally larger. While the overall sensitivity is decreased, the spatial specificity of the acquired signal is increased (De Martino et al., 2013b; Duong et al., 2003; Moerel et al., 2018a; Uludağ et al., 2009). Alternatively, functional imaging based on

cerebral blood volume (CBV) imaging using Vascular Space Occupancy (VASO) is possible (Huber et al., 2018a, 2014; Lu et al., 2003), which has demonstrated comparable specificity to that of 3D GRASE (Beckett et al., 2019). Mapping the location of large veins across the cortex, as possible with high spatial precision at UHF (Koopmans et al., 2008; Moerel et al., 2018a), may, in future work, allow examining and ultimately minimizing vascular artifacts that reduce spatial specificity (Box 2).

While the use of spin-echo fMRI at high magnetic fields (or other techniques with similar contrast weighting, e.g. 3D-GRASE or any technique requiring 180 degree transmit pulses) may be desirable as these contrasts provide increased specificity, these techniques rely on the use of $180°$ refocusing radio frequency pulses and therefore require higher power. In combination with the inefficient and inhomogeneous radio frequency transmission at UHF, these approaches can be rapidly limited by the specific absorption rate (SAR; i.e., the radio frequency power absorbed per unit of mass of tissue). Solutions to this issue are based on the use of local (single channel) surface coils or the use of multi-channel transmit coils in combination with methods to adjust the transmit field to the targeted areas (e.g. $B_1$ shimming or full parallel transmission) (see e.g. De Martino et al., 2012 for an application of $B_1$ shimming to the imaging of auditory cortical areas).

Another issue concerns geometric distortions and signal dropouts caused by large magnetic field ($B_0$) inhomogeneities that, in conventional EPI readouts, become much more prominent at ultra-high field. These problems are especially clear in the proximity of the ear canal and thus affect the lower portion of auditory cortical areas. Accurate $B_0$ shimming can alleviate, but not remove, these issues. In addition, parallel imaging (Griswold et al., 2002; Pruessmann et al., 1999) can be used to shorten the readout time and reduce geometric distortions as well as signal loss due to such high susceptibility regions. However, the use of parallel imaging approaches comes at the cost of signal to noise and at conventional acceleration factors (that will depend on the resolution of the images) dropouts are not completely removed from the acquired data. Readout times can also be reduced by reducing the field of view, by e.g. using outer volume suppression (Luo et al., 2001) or inner volume excitation (Oshio and Feinberg, 1991). This can be particularly helpful when trying to achieve very high spatial resolutions (Heidemann et al., 2012) in order to manage distortions. Many of these approaches (i.e., the use of single transmit coils for optimized transmit efficiency and outer volume suppression for minimizing field inhomogeneity induced artifacts) were used in the first auditory application at 7 T (Formisano et al., 2003).

High field scanners are becoming more readily available (De Martino et al., 2016), and auditory applications are increasing steadily. Compared to early work (Formisano et al., 2003), the transition from 3 T to 7 T has become much easier. Tested imaging sequences and protocols for both functional and anatomical imaging are readily available. The choice of the acquisition technique (sparse, clustered, continuous, or ISSS) follows the same considerations as those followed at 3 T, amounting to the trade-off between the biasing influence of scanner noise compared to the number of collected data points. The choice of imaging sequences for studying the auditory cortex is dominated by trade-off in SNR, coverage, temporal resolution, and spatial specificity, as detailed in Table 1. Imaging of subcortical auditory structures necessitates obtaining images with high spatial resolution.

Combined with the low SNR in central brain regions, this currently limits the choice to GE-EPI acquisitions for subcortical auditory imaging (Box 3).

## 3. Advancing auditory neuroscience through ultra-high field MRI

As a result of methodological advances, recent years have seen a rapid increase in auditory applications of ultra-high field MRI. These applications often take advantage of the available improvements in spatial resolution, to either image the small subcortical auditory structures, explore diverse MR contrasts with high spatial specificity, or obtain unique signals throughout the cortical depth. Below we review how UHF applications to auditory neuroscience have contributed to the current understanding of how sounds are represented throughout the network of auditory brain regions, and discuss potential future steps.

### 3.1. What sound processing takes place in human subcortical auditory regions?

The auditory pathway comprises multiple subcortical processing stages between the cochlea and the auditory cortex. Therein it differs from the visual system, where retinal input reaches primary visual cortex through only one relay station (the lateral geniculate nucleus in the thalamus). The auditory pathway, instead, includes the cochlear nucleus (CN), superior olivary complex (SOC), lateral lemniscus (LL), inferior colliculus (IC), and medial geniculate body (MGB) in the thalamus. As these subcortical auditory nuclei are small, and many of them contain even smaller functionally distinct subnuclei, their investigation in vivo is challenging. Size estimates of the subcortical auditory structures vary. Ex vivo estimates are considerably lower (CN = 46 mm$^3$, SOC = 7 mm$^3$, IC = 65 mm$^3$, and MGB = 58 mm$^3$; Glendenning and Masterton, 1998) than in vivo probabilistic estimates (e.g., CN = 24 mm$^3$, SOC = 63 mm$^3$, IC = 189 mm$^3$, and MGB = 207 mm$^3$ based on significant responses to sounds in at least 4/10 participants; Sitek et al., 2019). While brainstem-evoked responses using electroencephalography (EEG) and magnetoencephalography (MEG) and structural, functional, and, diffusion (dMRI) MRI studies contributed to auditory brainstem and midbrain research, much of our knowledge of the neurophysiology of these regions comes from animal and ex vivo human studies (Kandler, 2019; Schreiner and Winer, 2005). We know, for example, that neurons throughout the ascending auditory pathway respond best to a specific sound frequency and that neurons with a similar preferred frequency cluster together creating a tonotopic organization (CN: Rose et al., 1959; Ryugo and Parks, 2003, SOC: Guinan et al., 1972; Tsuchitani and Boudreau, 1966, LL: Bajo et al., 1999; Malmierca et al., 1998, IC: Merzenich and Reid, 1974, and MGB: Aitkin and Webster, 1972; Hackett et al., 2011; Imig and Morel, 1985). In addition to this general organizational principle, each processing stage in the auditory hierarchy displays its own distinct characteristics. The CN consists of partitions that differ in cell morphology, response characteristics, and connectivity. These partitions have been differentially implicated in the detection of timing cues (Tollin, 2003), cues relevant to sound localization in the vertical plane (dorsal CN - Oertel and Young, 2004; Reiss and Young, 2005), and the extraction of pitch and intensity information from complex sounds (Blackburn and Sachs, 1990; Kim et al., 1986). The superior olive is the first auditory region with confluent information from both ears and has been prominently implicated in sound localization based on both interaural time and level differences (Joris et al., 1998). The inferior colliculus regulates information flow from

the brainstem nuclei to the thalamus and cortex. It consists of three functionally distinct subdivisions. The lemniscal central nucleus has been shown to be involved in a variety of functions, such as the encoding of location and the representation of spectrotemporal acoustic cues (Aitkin and Martin, 1987; Ehret and Merzenich, 1988; Schreiner and Langner, 1988; Versnel et al., 2009). The non-lemniscal lateral/external and dorsal subdivisions have been implicated in multisensory processing and processing of feedback information, respectively (Aitkin et al., 1978; Winer et al., 1998). The MGB in the thalamus consists of three subdivisions (ventral, dorsal, and medial). Following IC processing, neurons in its lemniscal part (ventral MGB) were shown to act as spectrotemporal modulation filters. However, compared to IC processing MGB responses seem to be more strongly influenced by cognitive (i.e. task) and affective processing (Selezneva et al., 2017). This may be mediated by the non-lemniscal medial MGB subdivision, which is densely connected both within and outside the auditory pathway (including with multisensory targets and the amygdala; LeDoux et al., 1991). The MGB does not simply relay information to the cortex, but instead regulates the information flow (Antunes and Malmierca, 2011; Bartlett and Wang, 2007; Winer et al., 2005).

Despite being challenged by the effect of physiological noise (Guimaraes et al., 1998; Sigalovsky and Melcher, 2006), the possibility of measuring functional responses non-invasively from deep inside the brain with functional MRI has inspired early investigations of subcortical auditory processing. These early studies showed that functional responses can be reliably detected in the subcortical nuclei (Griffiths et al., 2001; Guimaraes et al., 1998; Harms and Melcher, 2002; Hawley et al., 2005; Sigalovsky and Melcher, 2006). The higher sensitivity available at ultra-high field UHF has allowed increasing the spatial resolution of functional and anatomical data to 1.1mm isotropic and below one millimeter, respectively. While 3 T allowed the evaluation of subcortical responses at group level, at UHF the reliability of subcortical responses in individuals could be evaluated (Fig. 2A). Specifically, comparing in vivo measurements with post-mortem cytoarchitecture and post-mortem MRI allowed evaluating the accuracy in identifying auditory subcortical regions both anatomically (García-Gomar et al., 2019) and functionally (Sitek et al., 2019). The results were encouraging, as they indicated that functionally defined regions were consistent both within and across individuals. Anatomical in vivo measures can aid in the identification of some of the nuclei (IC and MGB: Moerel et al., 2015; Sitek et al., 2019; Tourdias et al., 2014; SOC: Garcia-Gomar et al., 2019). Nevertheless, some localization discrepancies between in vivo MRI and post-mortem data were evident at the lower auditory processing stages (CN and SOC). The small size and proximity of the CN and SOC highlighted the importance of gathering high resolution data for a correct localization in individual brains. Future work may be directed towards understanding the nature of the neurovascular coupling in these regions, as that may explain some of the discrepancies with post-mortem data (Sitek et al., 2019). For example, taking advantage of the increased spatiotemporal resolution at UHF, a recent study showed that subcortical visual structures have a more rapid hemodynamic response function (HRF) than the primary visual cortex (Lewis et al., 2018). Similar investigations to detail potential HRF differences throughout the auditory pathway would be valuable. Future endeavors may further target the identification of anatomical contrast that can aid the individual localization of the CN, and explore the possibility to

separate ascending from descending anatomical subcortical pathways with dMRI as they target different subdivisions of the nuclei.

Beyond localizing subcortical regions, fMRI was used to explore their functionality. At 3 T, fMRI studies showed evidence for the relation between the average response in CN, SOC, IC, and MGB and various sound features (e.g. level, bandwidth, temporal structure, and repetition rate; Griffiths et al., 2001; Guimaraes et al., 1998; Harms and Melcher, 2002; Hawley et al., 2005; Sigalovsky and Melcher, 2006). Instead, the higher resolution afforded by high field functional imaging has allowed mapping the functional preference within subcortical regions - and their subnuclei – in vivo. That is, tonotopic maps were observed in the human IC and MGB (De Martino et al., 2013a,b; Moerel et al., 2015 ; Fig. 2B-C). Specifically, in the IC one frequency gradient was observed that was oriented from low-to-high frequency preference in dorsolateral to ventromedial direction. The MGB was shown to contain a low-high-low frequency gradient (running in dorsomedial to ventrolateral direction), and this allowed the definition of ventral MGB as well as a dorsally-located MGB sub-region (Moerel et al., 2015). Future work at UHF is needed to further characterize sound processing in human IC and MGB. Currently, all methodological requirements are met for this work to be performed. After detailing the processing of acoustic features other than sound frequency (e.g., spectrotemporal modulation tuning), it will be of particular interest to study responses in IC and MGB (and their subdivisions) in behaviorally relevant auditory settings. Recent studies have started this endeavor, and showed a differential contribution of dorsal and ventral MGB to speech recognition (Mihai et al., 2019). In addition, the ability to resolve tonotopic responses in the IC has allowed investigating their (frequency-specific) modulation with attention (Riecke et al., 2018). Future studies will hopefully continue this effort.

Due to their location, small size, and close proximity to each other, the functional exploration of the CN and SOC will be more challenging than that of the hierarchically higher auditory subcortical structures. However, given the localization of these regions based on both functional (Sitek et al., 2019) and anatomical (García-Gomar et al., 2019) group data, these structures can now be explored. Exploring the role of CN and SOC in acoustic feature processing of localization cues as well as information for complex sounds would be highly relevant. Ultimately, it would be of interest to functionally image the entire auditory pathway (from the CN to the auditory cortex), in order to study how sound representation evolves throughout the auditory pathway and how this representation is modified with task, learning, and disease. The large coverage afforded by most imaging approaches make UHF MRI an idea tool for this endeavor.

### 3.2. Where are human auditory cortical regions located?

Compared to the small subcortical auditory structures, the auditory cortex is much more accessible to non-invasive imaging methods. Accordingly, the cortical sound representation has been extensively studied with electrophysiology and 3 T MRI (e.g. Formisano et al., 2015; Poeppel and Hickok, 2015). However, in spite of decades of research, the location of human primary and non-primary auditory cortical fields is still under debate (Moerel et al., 2014). This is in sharp contrast with visual neuroscience, where a relatively short

retinotopic localizer is sufficient to identify both the location of the early visual cortical fields in individual hemispheres and their feature preference (Sereno et al., 1995). The inability to parcellate the human auditory cortex severely hampers the investigation of how the sound representation evolves throughout the auditory cortical hierarchy. The reason for the continuing debate in auditory neuroscience originates from several field-specific difficulties. First, even detailed histological parcellations of human auditory cortex vary considerably from each other (Galaburda and Sanides, 1980; Hackett et al., 2001; Morosan et al., 2001; Rivier and Clarke, 1997; Von Economo and Horn, 1930; Wallace et al., 2002; Zachlod et al., 2020). As a result, there is currently no 'ground truth' parcellation of human auditory cortex. Second, the macroanatomy of the human auditory cortex is highly variable (Heschl, 1878; Pfeifer, 1936, 1921; Rademacher et al., 2002, 1993; Steinmetz et al., 1989). Heschl's gyrus, the putative location of human primary auditory cortex (PAC), may be duplicated or even triplicated in individual brains and many individuals display intermediate macroanatomical variations (Benner et al., 2017; Marie et al., 2016, 2015; Zoellner et al., 2019). It is still unclear how microanatomy, and therefore functional specialization, relates to this macroanatomical variation. The macroanatomical alignment of an individual hemisphere to a microanatomical atlas of human auditory cortex is therefore problematic (Gulban et al., 2020). Third, maps of only one acoustic feature, frequency, are reliably reproduced across research labs (Ahveninen et al., 2016; Besle et al., 2019; Da Costa et al., 2011; Formisano et al., 2003; Moerel et al., 2012; Saenz and Langers, 2014; Striem-Amit et al., 2011; Woods et al., 2010). Tonotopic maps (i.e., maps of frequency preference), however, do not reverse at the border between PAC and non-primary auditory cortex, and therefore do not allow for the non-invasive definition of the PAC in individual hemispheres (Moerel et al., 2014). In comparison with the visual cortex, where primary areas are well conserved across subjects and normative databases with large sample sizes are available (Benson et al., 2018), establishing the parcellation and the variability of the PAC remains a challenge (Ren et al., 2020).

UHF MRI can help with all three difficulties. Anatomical imaging at UHF has the potential to provide a 'ground truth' parcellation of human auditory cortex, as it can be used to assess histological features of the cerebral cortex. For example, as the PAC is more densely myelinated than surrounding tissue due to dense thalamic projections (Nieuwenhuys, 2013), it should be identifiable based on myelin-related MR contrast. In accordance, Wallace et al. (2016) first showed a good correspondence between the density of myelinated fibers as assessed by myelin staining and the quantitative $T_2$* and $T_1$ weighted values measured ex vivo at 7 T. Next, they reliably identified PAC as well as several adjacent non-primary auditory fields based on MR images in a single human brain (Wallace et al., 2016, 2002). Repeating this study in a larger sample of hemispheres that vary in temporal lobe macroanatomy (i.e., by including several examples of all possible HG configurations) may allow detailing how the location of PAC relates to macroanatomical variation. Using myelin-related contrast, recent UHF studies identified PAC in vivo as well (Cohen-Adad et al., 2012; De Martino et al., 2015c). While PAC identification based on cortical variations in myelin density was also achieved at 3 T (Dick et al., 2012, 2017), the higher spatial resolution and contrast at 7 T allowed examining the laminar pattern of myelin density in the temporal lobe. Based on the cortical depth-dependent variations in myelin, various auditory

fields were defined through data-driven clustering, which reduced arbitrariness in setting a border (De Martino et al., 2015c).

While promising, myelin-related contrasts may not allow for the identification of all areas in the temporal lobe (Wallace et al., 2016). Extending myelin-based mapping with additional anatomical contrasts may help in the quest for a parcellation of the temporal lobe. For example, diffusion weighted MRI (dMRI) allows examining the orientation of fibers in the cortical gray matter. Comparing radiality across the cortex and throughout cortical depth (where high radiality means that cortical fibers run orthogonal to the white matter [WM]/GM border, while low radiality means that fibers run parallel to the WM/GM border) showed that the medial part of HG displayed high radiality throughout cortical depth (Gulban et al., 2018; Fig. 3). This microanatomical signature matches the known fiber orientation of PAC based on myelin staining (Nieuwenhuys, 2013). Furthermore, as the thalamocortical connectivity varies between auditory cortical fields (Winer et al., 2005), it would be of interest for future studies to use dMRI and tractography between the ventral MGB and the superior temporal lobe to parcellate the auditory cortex. UHF may also be of use for this endeavor, as previous work showed that spatial resolution as well as high angular resolution is beneficial to resolving fiber crossings (Roebroeck et al., 2008).

Additional information regarding auditory fields may be acquired through functional imaging. Recent years have seen the search, both at 3 T and 7 T, for an acoustic feature with a cortical representation that is orthogonal to the tonotopic map, as this would allow for auditory cortical parcellation. Results revealed an organized variation in frequency selectivity (tuning width) along the superior temporal plane, with more narrowly tuned neuronal populations along HG putatively identifying PAC (Moerel et al., 2012; Fig. 3). Furthermore, tuning to temporal modulations (i.e., the variation in sound energy over time) was suggested to be organized orthogonally to the tonotopic map (Barton et al., 2012), but was alternatively suggested to display a large-scale medial to lateral organization covering the superior temporal lobe (Herdener et al., 2013; Santoro et al., 2014). Parcellation of the auditory cortex was also achieved based on resting state functional connectivity, which identified the primary auditory regions by their dense connectivity to the MGB (Glasser et al., 2016). To date, none of these functionally-based options for the identification of auditory cortical fields have been widely adopted by the auditory community.

UHF fMRI may add to the ongoing discussion regarding the representation of acoustic features in auditory cortex, as this representation may change with spatial resolution (Gentile et al., 2017). Moreover, the increased spatial resolution at UHF allows examining the stability and variability of feature maps with cortical depth. The cortical depth-dependent examination of acoustic feature tuning has only been carried out by a handful of studies, using either GE-EPI (Ahveninen et al., 2016; Moerel et al., 2019) or 3D GRASE (De Martino et al., 2015b; Moerel et al., 2018a, 2018b). Common to these studies is a stable representation of frequency, as well as other acoustic features (i.e., spectral and temporal modulations) throughout cortical depth and observed over large parts of the auditory cortex (Ahveninen et al., 2016; De Martino et al., 2015b). However, cortical depth-dependent variation in acoustic feature preferences was present as well (Moerel et al., 2018b; Fig. 4). As expected, the GE-EPI data showed an improvement in tonotopic mapping after voxels

intersecting the pial surface were excluded (Ahveninen et al., 2016), reflecting the lower specificity towards the pial surface in GE-EPI data. 3D GRASE measurements showed stability in specificity throughout cortical depth (Moerel et al., 2018b). For future work, it would be of interest to explore the columnar stability of neuronal population tuning to higher-level acoustic features (such as pitch, timbre, and harmonics-to-noise-ratio [HNR]; Allen et al., 2017; De Angelis et al., 2017; Frühholz et al., 2016; Norman-Haignere et al., 2013). As in visual cortex, cortical depth-dependent stability in feature tuning may vary across the cortical surface (Nasr et al., 2016; Yacoub et al., 2008), and may thereby provide a source for the parcellation of the superior temporal plane based on functional responses.

### 3.3. How do sound category preferences result from acoustic feature processing?

Building on low-level acoustics as input, an abstract sound representation is created via auditory cortical processing (Da Costa et al., 2015; Giordano et al., 2013). Despite numerous investigations, how categorical responses emerge from the different acoustic features that serve as input to the early auditory cortex remains unclear. Studies at conventional field strengths (3 T) and low spatial resolution (> 2 mm) have shown that clusters along the superior temporal gyrus and sulcus (STG/STS) respond preferentially to sounds of several categories, most notably speech (Binder et al., 1994), vocalizations (Belin et al., 2000), and music (Norman-Haignere et al., 2015). Such preferential responses have been interpreted as evidence for functional specialization and category-specific processing, independent of acoustics. However, the extent to which acoustic features influence responses in categorical parts of the auditory cortex is still under debate. The debate is fueled by low-resolution (3 T) results showing that category-preferential responses are diminished when compared to sounds of different categories that are acoustically similar (Staeren et al., 2009) as well as that in these category preferential clusters, frequency selectivity can be assessed with sounds of non-preferred categories (including tones; Moerel et al., 2012). In addition, UHF fMRI (i.e. with higher sensitivity) has shown that in these categorical regions the response to the non-preferred sounds (e.g., music in a voice preferring region) allows reconstructing the acoustic content of the sounds (Santoro et al., 2017).

The key question is what computations underlie the emergence of functional specialization in categorical regions. Answering this question with fMRI requires the acquisition of brain responses to sounds and the use of tools to link algorithms to brain responses. Choices in acquisition and analysis techniques may influence, and possibly bias, the obtained results. In terms of acquisition, detailing the computations underlying categorical responses requires measurements that approach the spatial scale at which these computations are performed. As lower resolution fMRI data (and in particular in 3 T data) is biased towards the processing in superficial cortical layers (see section 2), it may overrepresent the output of intra-cortical computations or the effects of feedback to a cortical area. Instead, UHF fMRI allows to access to fundamental computational units of the brain. Because of the increased spatial resolution (and specificity) of the signal at UHF, computational neuroimaging approaches are particularly interesting when combined with measurements at high field strength (De Martino et al., 2016).

Recent years have seen the development of a number of tools that allow studying the computational principles of the human cortex (i.e., population receptive field mapping: Dumoulin and Wandell, 2008; encoding: Kay et al., 2008; Naselaris et al., 2011; representational similarity analysis: Kriegeskorte et al., 2008; Mur et al., 2008). We used encoding to show that the sounds' spectrotemporal modulation content (i.e., modulations in the sounds' energy over frequency and time) explained fMRI responses throughout the auditory cortex (Santoro et al., 2014). The sound's spectrotemporal modulation content explained responses to sounds significantly above chance in regions that included categorical regions in the STG/STS, suggesting that spectrotemporal modulations influence STG/STS responses. In fMRI encoding, natural stimuli (i.e., natural sounds) are used to determine the ability of a range of models to predict fMRI responses. While driving the cortex in an ecologically valid manner, the disadvantage of this approach is that it is difficult to disentangle the effects of low-level (i.e., acoustics) from higher-level features in natural stimuli. In fact, sounds from different categories also differ acoustically (Norman-Haignere and McDermott, 2018) to the point that sound categories can be determined on the basis of acoustics (almost perfectly for, e.g., speech and music). As a result, there may be a bias when assessing the relevance of acoustic processing when that is based on an acoustic model's prediction accuracy of responses to natural sounds. Tackling this bias, a recent study proposed to compare responses to natural sounds to the ones elicited by synthetic stimuli matched in acoustic content to naturals sounds (Norman-Haignere and McDermott, 2018). The rationale underlying this study was similar to classical cognitive-subtraction: a voxel mostly driven by acoustics (e.g., by spectrotemporal modulations) should show identical responses to natural sounds (NS) and model-matched (MM) synthetic sounds. As responses to MM and NS in the STG/STS were not significantly similar (bottom panel Fig. 5A), the authors concluded that spectrotemporal modulations do not influence the response in these areas (Norman-Haignere and McDermott, 2018). However, the absence of evidence should not be interpreted as evidence for absence. Doing so comes with an inherent false negative error (type-II; i.e., the acoustic model could still be driving responses in STG/STS) that can be quantified with simulations (Fig. 5B). While for PAC-like responses (left panel of Fig. 5B) the type-II error ranges around 10 %, in non-PAC it increases drastically (above 40 % - right panel of Fig. 5B).

These observations highlight that the application of advanced computational methods require scrutiny. A safer interpretation of the computational fMRI studies carried out so far across field strengths is that low-level acoustics influence STG/STS responses (Santoro et al., 2014), but that part of the response in these regions can be explained by a functional specialization that reaches beyond low-level acoustics (Norman-Haignere and McDermott, 2018). As a result of feedback signals as well as intra-cortical (across laminae; within a column) processing, the sound representation in superficial and deep layers may be more complex than in middle input layers (Hirsch and Martinez, 2006; Linden, 2003). In PAC, the increase in processing complexity through cortical layers has been shown both using electrophysiology in cats (Atencio et al., 2009) as well as in humans with UHF laminar fMRI (Moerel et al., 2019). In STG/STS, middle layers may partially represent the acoustic content while superficial layers could show an increasingly categorical response due to intra-cortical processing across layers (or feedback). The latter may be disproportionally

represented in low resolution data (and in particular in 3 T data). We expect that UHF fMRI at high spatial resolution will prove instrumental to better understand the processing in STS/STG by separating responses across cortical layers, and that this will resolve the debate on the relevance of acoustical processing in lateral temporal cortex (Norman-Haignere and McDermott, 2018; Santoro et al., 2014).

Processing in the auditory cortex is extremely flexible. Feature tuning rapidly adapts to changing task demands (Fritz et al., 2003; Yin et al., 2007), and attention-induced changes were shown to be cortical depth-dependent (De Martino et al., 2015b; O'Connell et al., 2014). Beyond the laminar mapping of tuning to low-level and more complex acoustic features, it would be of interest to explore the flexibility in feature processing when subjects engage in meaningful tasks. That is, future UHF studies may explore auditory cortical processing during changes in context, attention, and task performance.

## 4. Discussion

In sum, while technical challenges such as the loud noise of image acquisitions, magnetic field inhomogeneities, and limitations in RF transmit efficiency remain, UHF MRI has, in recent years, seen many successful applications to the investigation of the human auditory system. With the rising accessibility and ease-of-use of ultra-high field scanners, including a 7 T scanner approved for clinical use, the number of applications is expected to rapidly rise in the coming years. Here we have reviewed the use of UHF to study three unresolved challenges in the human auditory neuroscience literature. First, the increased spatial resolution of UHF fMRI has been used to explore processing in the small subcortical auditory structures and their subnuclei. The groundwork of exploring IC and MGB processing is complete, and processing in these structures within relevant task-settings can now be explored. Moving ahead, the exploration of the even less accessible CN and SOC in the human brain is now at its starting point. Second, signatures of distinct human auditory cortical regions can be gathered through diverse MR contrasts. The combination of myelin-related and diffusion-weighted anatomical images with functional contrasts at UHF holds great promise towards elucidating the relation between human auditory cortex micro- and macroanatomy, as well as to provide an in vivo auditory cortical parcellation at the level of individual subjects. Third, UHF fMRI was successfully used to study auditory cortical columnar computations. Applying these techniques to the investigation of higher-level acoustic features may reveal the sound transformations that underlie the emergence of functional specialization in higher-order cortical regions.

The increased spatial resolution of UHF data allows disentangling feedforward from feedback information as these signals are localized at different cortical depths (Douglas and Martin, 2004) and distinct sub-nuclei in subcortical structures (Winer et al., 1998). In the visual system, the spatial cortical organization has been exploited to study laminar differences between the influence of top-down and bottom up processing in general (Lawrence et al., 2019), and the influence of context on the stimulus representation (Kok et al., 2016; Muckli et al., 2015). While contextual processing is as relevant for audition as it is for vision (Heilbron and Chait, 2018), in the human auditory system grounding contextual processing on laminar computations (or subdivision of subcortical structures)

remains relatively unexplored. That is, to date in vivo explorations in the human auditory brain at sub-millimeter resolution focused mostly on broad task modulations (De Martino et al., 2015b; Riecke et al., 2018). The use of UHF fMRI to study auditory context thus represents a future challenge for auditory neuroscientists.

## Supplementary Material

Refer to Web version on PubMed Central for supplementary material.

## Acknowledgements

This work was supported by the Netherlands Organization for Scientific Research (NWO) [grant numbers 451-15-012 and 864-13-012]; the National Institutes of Health (NIH) [grant numbers P41 EB027061, P30 NS076408, and RF1 MH116978]; and the Dutch Province of Limburg.

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

**Box 1**

**Next steps in the UHF MRI exploration of the human auditory system.**

- Test silent functional imaging techniques such as SWIFT and evaluate their sensitivity and specificity

- Compare the sensitivity of acquisition techniques combined during fast temporal sampling

- Understand subcortical neurovascular coupling to explain discrepancies with post-mortem data

- Identify an anatomical contrast to localize the CN

- Anatomically separate ascending from descending subcortical pathways using dMRI

- Explore the role of CN and SOC in acoustic feature processing

- Obtain a 'ground truth' parcellation of human auditory cortex by assessing histological cortical features

- Explore the columnar stability of neuronal population tuning to higher-level acoustic features

- Study how the sound representation throughout the entire auditory pathway, from the CN to the auditory cortex, is modified with context, task, learning, and disease.

- Clinical applications

**Box 2**

## What is the neuroscientific value of parcellating the human auditory cortex?

Efforts to parcellate the human auditory cortex into distinct subfields based on in vivo measures are abundant (Barton et al., 2012; De Martino et al., 2015c; Dick et al., 2012; Moerel et al., 2012), yet the value of obtaining such a parcellation for auditory neuroscience may be questioned. That is, one may argue that a parcellation would not, necessarily, extend our knowledge on how auditory information is processed in the brain. Here we argue in favor of the value of obtaining a parcellation. Our rationale follows from the hypothesis that an accurate cortical parcellation identifies areas with unique functional properties. The ability to identify such areas across individuals (and studies) would allow merging results in a common reference space. Such an approach would substantially reduce the variability introduced by, e.g., the variable macroanatomy of the human temporal lobe. Without a common reference space, or using a reference space that does not correctly align functionally identical regions to each other (e.g., through the use of a probabilistic atlas), neuroscientific effects could be blurred or even removed. This is especially true for UHF MRI applications in which, due to the push for high spatial resolution, signal-to-noise ratio (SNR) is limited. In cortical depth dependent studies, for example, averaging (or spatially smoothing) data within functionally coherent areas (while maintaining separate the signal across depths) is a common practice used to increase SNR. This requires the area to be defined correctly defined to avoid averaging across regions with functionally distinct properties. The areal definition also needs to be performed in an unbiased manner in order to avoid double-dipping. While functional localizers (or the combination of functional and anatomical data) can be used to define some auditory regions (e.g., voice regions; (Belin et al., 2000), such localizers are not available for the majority of the auditory cortex. Obtaining a parcellation of human auditory cortex would allow the unbiased definition of areas. Furthermore, it would allow the assignment of homologies across species and thereby bridge results obtained using non-invasive measurements to the human brain.

**Box 3**

### Can laminar and columnar ultra-high field fMRI studies contribute neuroscientific knowledge?

Is it possible to draw entirely new neuroscience conclusions from laminar and columnar UHF fMRI studies, beyond the replication of what is already known from animal neurophysiology? This critical question is perhaps not surprising, as most of the early UHF fMRI work has focused on replicating known neuroscientific organizations (e.g., ocular dominance and orientation columns in primary visual cortex (Yacoub et al., 2007); tonotopic columns in primary auditory cortex (De Martino et al., 2015b)). Even to date, known neuroscientific organizations are studied in order to validate the employed methodology or evaluate strengths and weakness of novel UHF techniques (De Martino et al., 2013b; Huber et al., 2018b; Kemper et al., 2015; Moerel et al., 2018a). However, recent years have also seen applications beyond early cortical regions shedding light on the intra-cortical computations relevant to complex tasks (e.g. working memory; Finn et al., 2019). The main strength of ultra-high field fMRI lies in the combination of the spatial resolution required to access the mesoscopic scale (accessing feedforward and feedback processing through subdivisions of subcortical regions and cortical layers), while maintaining the coverage required to image a large part of brain. Thereby, this tool is optimally suited to gain an understanding of intra-areal interactions and processes. The ability to measure high spatial resolution signals with large coverage enables studying how information is transformed between brain regions. In auditory neuroscience, imaging the complete auditory pathway at a high spatial resolution allows investigating how sound representations are modified by task, learning, and disease. UHF fMRI is unique in this respect, and its results will surely provide novel neuroscientific insights. In particular, we expect the testing of theories that make specific assumptions regarding the mesoscopic implementation level of brain processes and/or rest on the hypothesis of hierarchical interactions across regions (e.g. predictive coding) in the near future.

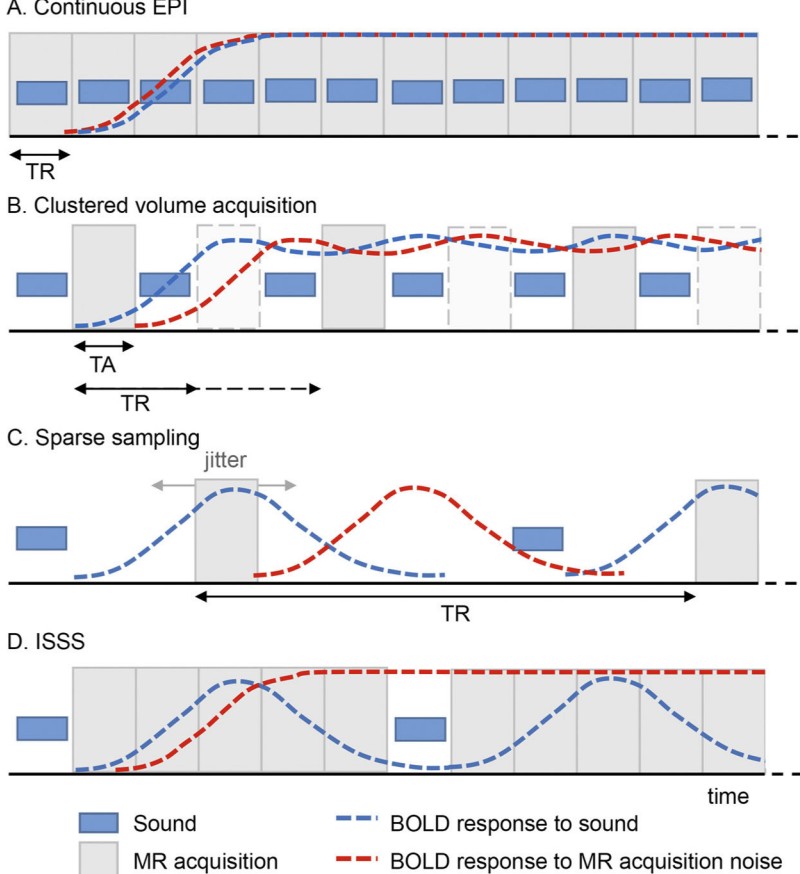

**Fig. 1. Auditory neuroimaging approaches.**
The blue boxes represent sound representation, and the gray boxes represent MR data acquisition. The blue and red dashed line represents the BOLD response evoked by the sounds and the MR acquisition noise, respectively. The various approaches represent a trade-off between (lack of) BOLD signal contamination and experimental power, with continuous EPI (A) and sparse imaging (C) reflecting the extremes, and clustered volume acquisition and interleaved silent steady state (ISSS; B-D) representing compromises. TA = acquisition time; TR = repetition time.

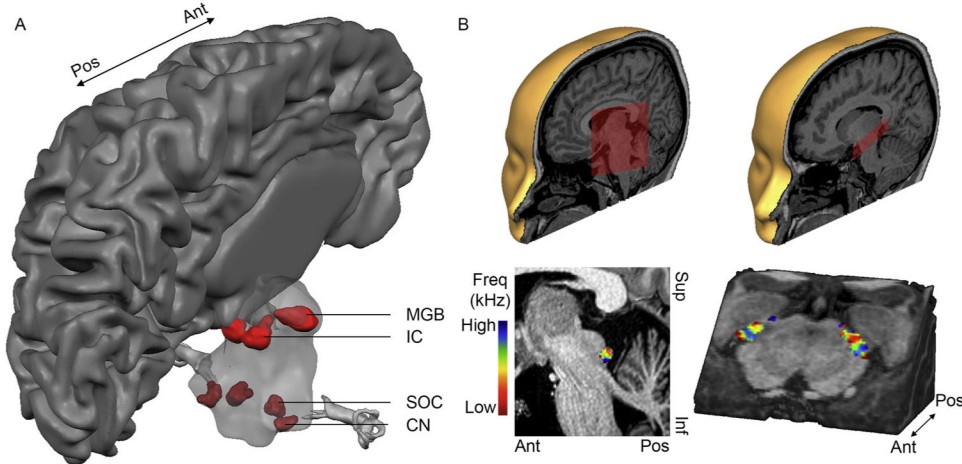

**Fig. 2. Ultra-high field imaging of the auditory pathway.**
(A) Using functional MRI at 7 T, auditory functional responses can be reliably measured throughout the auditory pathway in an individual brain. CN = cochlear nucleus, SOC = superior olivary nucleus, IC = inferior colliculus, MGB = medial geniculate nucleus, AC = auditory cortex. (B–C) Group tonotopy maps, where low and high frequencies are shown in warm and cool colors, respectively. (B) One tonotopic gradient can be observed in human IC, running from low-to-high frequency preference in dorsolateral to ventromedial direction. (C) A mirror-symmetric low-high-low frequency gradient is present in human MGB, running in dorsomedial to ventrolateral direction. Figure panel A is adapted from Gulban et al. (2016). Figure panels B–C are adapted from Moerel et al. (2015).

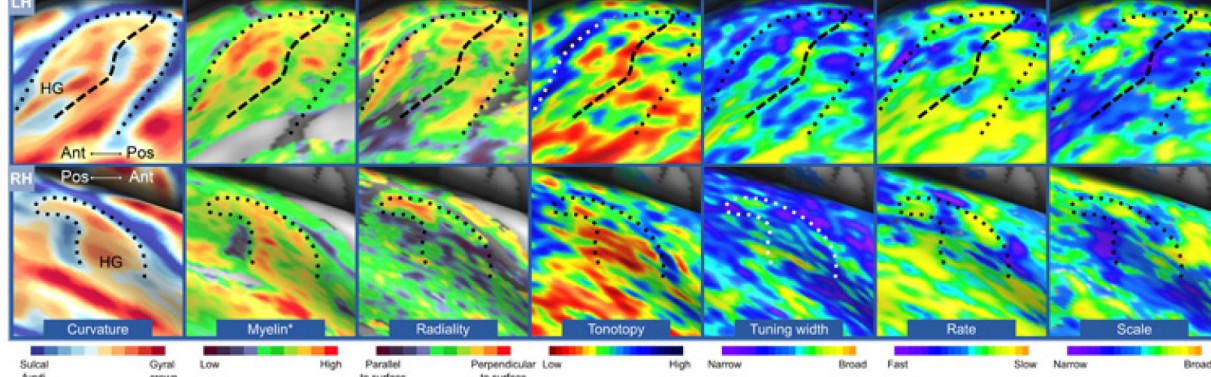

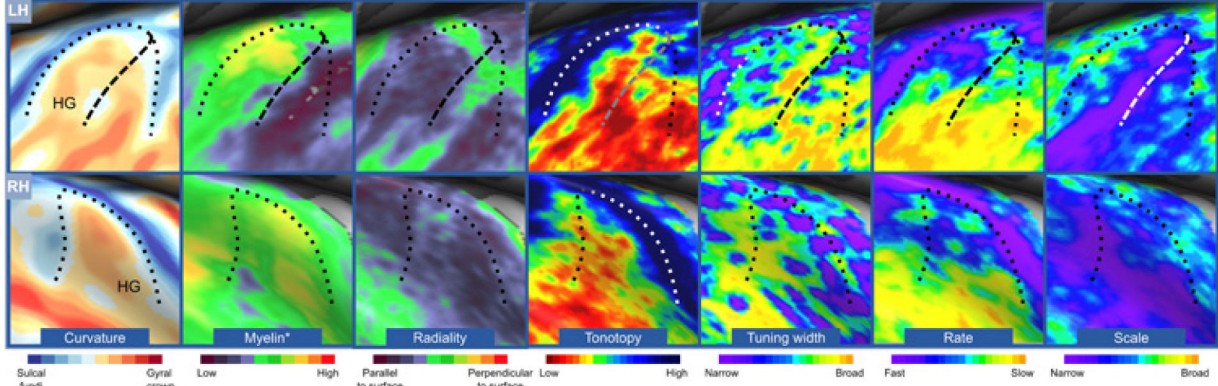

**Fig. 3. Identification of PAC using various MR contrasts.**
The upper and lower panel show an individual participant and group data ($N = 10$), respectively. The anteromedial part of Heschl's gyrus (HG), putative PAC, is functionally characterized by a complete tonotopic gradient, narrow tuning width, fast temporal modulation rates, and broad spectral modulation scales. Myelination is high in this region, and radiality is greater (i.e., more perpendicular to the WM–GM surface) in the medial part of HG than the lateral part. The dashed and dotted black line identify the outline of HG and the intermediate sulcus, respectively. Figure from Gulban et al. (2017).

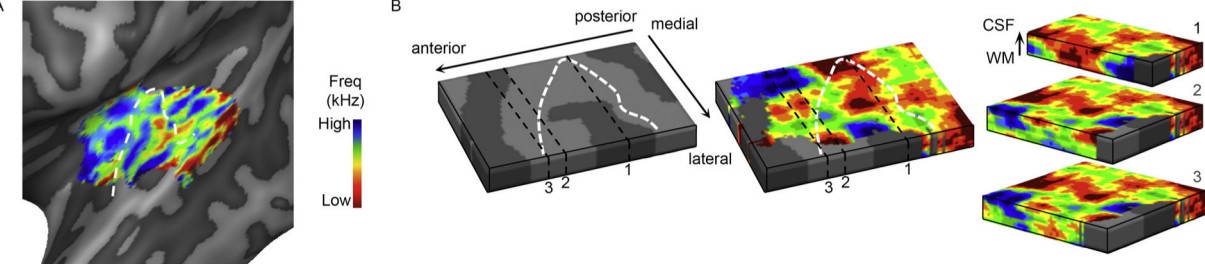

**Fig. 4. Cortical depth-dependence of frequency preference.**

Frequency preference, as mapped using 3D GRASE, is shown in the left hemisphere of an individual volunteer. The tonotopic map is shown both at macroscopic level (A) and after zooming in to show frequency preference throughout cortical depth (B). The cortical depth-dependent maps show regions of stable frequency preference (i.e. frequency columns) as well as variations in preference throughout depth. The white dotted line follows the crown of HG (a partial duplication, resulting in 1.5 H G, is present in this hemisphere). Figure adapted from Moerel et al. (2018b).

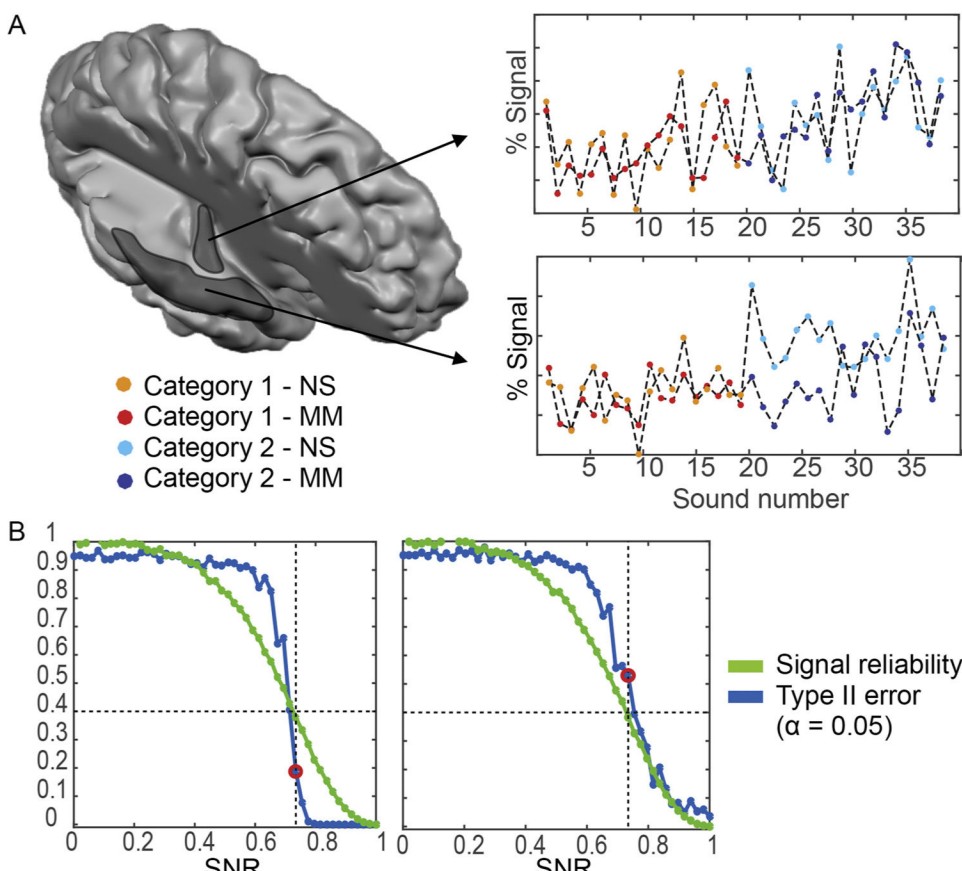

**Fig. 5. Simulation results.**

A) Simulated fMRI responses to natural sounds (NS) and model matched sounds (MM) for a representative voxel in putative primary and non-primary auditory cortical regions (PAC and non-PAC), respectively, and the associated noise corrected normalized squared error (ncNSE). The ncNSE is a measure of the similarity between two response vectors (i.e., the similarity between the response to a set of NS and the corresponding MM sounds). In this simulation, sounds are assumed to belong to two categories (e.g., voices and non voices). The simulations are built so that a voxel's fMRI response is always driven by the acoustic content of the sounds, while the influence of the sound category differs between cortical regions (PAC vs. non-PAC) and the type of sounds (NS vs. MM). In particular the categorical influence in PAC is small and equal between NS and MM sounds. Instead, in non-PAC NS sounds are characterized by a larger contribution of the sound category. For these simulated responses, concluding that there is no influence of acoustics based on the ncNSE therefore represents a false negative. B) For different levels of SNR, the proportion of false negatives (type-II error; blue curve) associated to an α-level of 0.05 obtained after 5000 simulations, together with the data reliability (green curve) as measured in Norman-Haignere and McDermott (2018). The left and right panel report the proportion of false negatives in the putative PAC and non-PAC scenario, respectively (i.e. in case of an equal vs. different influence of categorical information between NS and MM sounds). The

proportion of false negatives associated with a data reliability of 0.4 is highlighted by the red circle.

**Table 1**

Strengths and weaknesses of fMRI sequences at 7 T.

|  | GE-EPI | 3D GRASE | VASO |
| --- | --- | --- | --- |
| SNR | + + | − | + |
| Coverage | + + | − | + |
| Temporal resolution | + + | + + | − |
| Spatial specificity | − | + + | + + |

*Prog Neurobiol.* Author manuscript; available in PMC 2022 March 01.