## [Peer Review File · Progress in neurobiology]

Peer Review Overview

Manuscript Title: “Using high spatial resolution fMRI to understand representation in the auditory network”

Received	30-Jan-2020
1st Decision	11-Apr-2020
Revision Submitted	27-May-2020
Accepted	13-Jul-2020

Decision Letter

Ref.: Ms. No. PRONEU-D-20-00047

Title: Using high spatial resolution fMRI to understand representation in the auditory network

Dear Dr. De Martino,

Thank you for submitting your manuscript to Progress in Neurobiology. We have received comments from reviewers on your manuscript. Your paper should become acceptable for publication pending suitable minor revision and modification of the article in light of the appended reviewer comments.

When resubmitting your manuscript, please carefully consider all issues mentioned in the reviewers' comments, outline every change made point by point, and provide suitable rebuttals for any comments not addressed.

Please resubmit your manuscript by Jun 10, 2020.

We look forward to receiving your revised manuscript.

Kind regards,

Luca Vizioli
Guest Editor

Sabine Kastner
Editor-in-Chief
Progress in Neurobiology

Comments from the Editors and Reviewers:

Reviewer 1

Very interesting paper. I don't have major comments apart from suggestions for increasing the focus, or minor editing remarks.

- The review of the sparse sampling sequences is very interesting. I wonder whether it might help the reader to have some graphical representation of the advantage of sparse sampling vs. e.g., a noisy EPI sequence, but with a very fine temporal sampling (and more data points for a GLM-based estimation of BOLD responses). If noise is detrimental even when the fMRI time series is sampled very finely in time, then one should see clear differences in this comparison.
- The review of UHF applications to the imaging of subcortical structure of the auditory system is remarkable and very well justified.
- Part of the review of the category vs. acoustics studies is about limitations in variance partitioning methods and in model-matching methods. I do love these considerations, and I think they are important. However, I think they start decreasing the focus of the manuscript from the main topic: how does UHF help auditory neuroscience? A better justification for these considerations might be appropriate. Again, most of the section is about non UHF literature, and the majority of UHF-related considerations are in large part about future research directions.

Detailed remarks:

- P[age]#,L[ine]# (including headings etc.): [original text] proposed edit/comment
- P3, L16: [small subcortical structures] give example, and approximate volume estimate?
- P3, L16: [inner workings] a bit of an approximate description. What about "that could not be resolved in detail at lower fields"?
- P3, L27: [In addition... entire paragraph] the usefulness of these improvements in UHF anatomical imaging to auditory neuroscience are not crystal clear. Could you elaborate?
- P4, L12: [uniquely human traits (i.e., speech and music)] speech and music are not traits. In general, I find this entire sentence a bit unnecessary: non-invasive exploration of the auditory system is not limited to UHF imaging. And I am not sure you need to convince the readership of your paper that audition is important. They wouldn't open it if they weren't convinced already ;-)
- P4, L12: [driven by these unique demands... rest of sentence] speech is unique to humans but vocal communication isn't. In general, reducing the importance of hearing to speech and music is an overstatement. Otherwise we could cross a road while wearing a muffler and still survive, or walk on glasses that were just broken out of your sight (but not out of our hearing) and still not hurt ourselves... Again, I am not sure it is important to convince readers here that audition is important. Finally, if we consider hearing as object processing through a non-visual modality (that goes beyond corners like smell but no other), it becomes clear that important hearing abilities are shared throughout much of the animal kingdom.
- P4, L18: [Here, we review the use of UHF MRI for studying the human auditory system] Here, we review auditory neuroscience studies based on UHF MRI.
- P4, L21: [First,... rest of paragraph] The introduction is slightly redundant, surely starting from here. I would personally think that making this points briefly right away in the first paragraph of the MS (along with some shrinking of the subsequent content, if possible) might further streamline the introduction.
- P4, L22: [enabling studying how sounds are processed] enabling the study of sound processing
- P4, L25: [may be localized individual hemispheres] I would try to be more specific here: someone naive to UHF imaging might think that it's already possible to localize cortical regions on each hemisphere, at least approximately, using e.g., anatomical atlases.

- P4, L26: [study the columnar computations] "shed light on column specific computations" would be more acceptable. We don't study computations directly, particularly with fMRI (regardless of the field strength).
- P4, L27: [This may show an abstract] not really something that can be done with UHF fMRI only.
- P4, L29: [understanding] mechanistic understanding
- P5, L5: [so do the gradients move] do gradients actually move? Or is something else whose movement is transformed into acoustical energy?
- P5, L10: [in which the forces were compensated locally] some more specifics would be interesting given the importance of the issue.
- P5, L12: [attenuating the scanning noise... Sensimetrics] does Sensimetrics use active noise cancellation? If not, how is noise attenuation achieved?
- P5, L23: [which has] and has
- P5, L23: [advantage of fully separating the BOLD response to the sound from the BOLD response to the scanner noise] this would be true if e.g., one estimated the BOLD response to the scanner noise and the BOLD response to each sound stimulus with HRF-convolved regressors for each. Is this usually the case in HRF fMRI for auditory neurosciences? Also, you might specify "sound stimulus" instead of "sound" only.
- P5, L26: [severely limiting the power of the experimental design] or severely increasing the uncertainty of GLM-based (etc.) BOLD estimates because of the reduced number of data points?
- P6, L15: [on the order] in the order
- P6, L29: [https:...] is this the right citation format for this resource?
- P7, L10: [necessitate] require
- P7, L12: [specific absorption rate] a definition would help
- P7, L15: [et al. [2012] for] et al., 2012, for
- P7, L17: [Furthermore] not a great way to start a new paragraph. An alternative such as "another issue" or similar would be better
- P11, L16: [(communication)] why this particular class of sounds?
- P13, L15: [Alternatively,] similar remark as above for paragraph starting with "Furthermore". Perhaps remove "alternatively".
- P16, L10: [Second, the principles...] might be worth it to explicitly state that this regularization is required to do e.g., encoding analyses, whereas OLS variance partitioning is fine in the absence of regularization requirements.
- P16, L14: [their performance under crossvalidation] this is again specific to encoding. Cross-validated variance partitioning in the absence of regularization should be much more straightforward...
- P18, L16: [these include pitch, timbre and harmonics-to-noise-ratio] I believe other fMRI studies not mentioned here, and predating the majority of those in these list, investigated all these features at once, and some more as well.

Reviewer 2

Summary:

In this work, the authors review the state-of-the-art of ultra-high field MRI for studying the human auditory system non-invasively. The manuscript is well-written and organized, and represents a comprehensive overview from researchers who have been responsible for many key studies that now make the technique more accessible to other researchers. It will likely serve as an important reference for auditory researchers in the coming years.

My comments are relatively minor and mainly consist of suggestions for flow and consistency, see below.

Best regards,
Emily Coffey

Comments:

1. Part of the purpose of this work seems to be to encourage a larger segment of the auditory neuroscience community to adopt 7T for a subset of research questions for which it is uniquely suited. While the explanation of the opportunities are very convincing, this 3T reader started to feel in need of some reassurance concerning the accessibility of the technique for wider application in auditory cognitive work, specifically around page 7. It could be very helpful for users to have something like a "best practices" "starting point recommendations" or decision tree diagram (maybe for a few common designs?), inset box or paragraph to provide some advice as to where to get started making experimental / scan parameter decisions. Failing that, perhaps at least a sentence or two to the effect that while many of these matters need further study, we are at the point where some design choices are known/semi-optimized such that cognitive work can proceed.
2. I suggest to include a brief overview of the organization of the rest of the paper at the end of the introduction to orient readers.
3. Pg. 8, concluding paragraph to section 2 was about solutions for scanner noise - I think this would make sense placed after the paragraph about scanner noise being the main challenge for auditory UHF. The point about HF driving technology didn't come across very clearly for me and I don't think is a critical one, so could be removed. The sentence that starts with "Despite the improvements (...)" also seemed to be setting up for an argument that didn't really happen, could be reworded to just emphasize that things are pretty good but will get better, and here are some of the ways.
4. Pg. 9, subcortical structure overview: "The partitions of the CNs (...)" - this long sentence is hard to parse as the structure of the phrases is not parallel, suggest to highlight function and structure in the same order and form for each. The sound localization / SO point is also introduced twice with an intervening point about CN, perhaps just reorganize them according to ascending order of structures. IC is introduced as an "obligatory relay station". The authors clearly don't mean that it only does that, and specifically make the point for the MGB that it's not only a relay just a few lines after, but as some of the auditory subcortical people have been fighting against the idea that the IC is just a relay, it could be better to use another word. "regulates information flow" is intriguing but vague, perhaps an example or two of what that means from those papers?
5. Pg. 13 "To date, none of these functionally-based options (...) has been adopted by the community" - why not? As an auditory researcher I sometimes struggle to decide which of the many parcellations to use, so I am intrigued to know why the authors think the poor adoption is the case, and it would better help me understand why UHF could help add to the discussion, which is the subject of the subsequent paragraph. Do they authors think this is because they are not seen to be reproducible, vary too much by analysis technique, too hard to implement, or other factors?

6. Pg. 14 the section about UHF options that differ in sensitivity seemed more similar to the challenges/methods material in section 2, as it is quite methodsy and might distract from the focus of section 3. It could be moved to section 2, perhaps renaming that "Auditory specific methodological considerations" (to include both challenges and positive points), and then the text in 3 could focus more on the neuroscience questions and quickly refer to the fact that there are scanning sequences that can get at those questions. However, this is only a suggestion that would appeal to me organizationally and I leave it to the authors.
7. P.11 + The authors have some excellent suggestions for next steps, but they are scattered around the end of the manuscript. If the journal format allows, it could be nice to also collect them into a point-form Box, similar to how TiCS papers are organized.
8. P. 20 The final paragraph (about clinical advances) seems a bit tacked on there and is not related directly to the title and main focus of the paper. Though it is nice to include, I would suggest to poke that material in somewhere else, maybe even in the bottom of the introduction and then say that while clinical advances are promising and increasing, you're going to focus on understanding representation (i.e. basic science) in the rest of the paper. The last sentence in the penultimate paragraph is a nice strong positive conclusion about UHF and auditory neuroscience.

Very minor:

- Use of the word "Interestingly," seems slightly odd contextually in two places, on pg. 6 and 7 - nothing is lost by its removal
- Check consistency of "in vivo" vs. "in-vivo", and I believe depth-dependent should be hyphenated, though it may be a stylistic choice
- Pg. 6 "An exception to this is represented" - due to length of preceding sentence, this reader had to check what the referent was, could make clearer
- Pg. 9. "As the IC," - word missing? "As in the IC," or "Analogous to the IC"?
- Pg. 18. "This may resolve conflicting results across previous studies" - (e.g. REFs) would be helpful here

Reviewer 3

This is a high-quality review of the potential applications of ultra-high field (UHF) anatomical and functional MRI in studies of human auditory system, a domain that could benefit from UHF imaging even more than research on the much more intensively studied visual system. The authors offer interesting perspectives on how to deal with obstacles of using UHF fMRI in studies of the auditory system, such as the more general problem of acoustical noise, as well as the more specific challenges such as deficiencies in transmit B1 profiles in or near the superior temporal plane. On the other hand, the manuscript communicated many new ideas and guidance on research areas where future UHF studies could be most fruitful, such as the functional mapping of subcortical auditory pathways and the functional parcellation of human auditory cortices. This review will likely be very well received by the human auditory neuroimaging community. I do not have any major concerns or objections, but I have listed a few more specific suggestions below.

Specific comments:

The manuscript could be made even more interesting and useful for those who are just entering the field of UHF MRI of auditory system by making it more assertive or opinionated. For example, the authors could more clearly express their assessment of available scanning

approaches, including conventional continuous, sparse sampling, clustered sampling, and ISSS, as well as of the potential trade-offs using approaches such as SWIFT. How about smoothing?

The section 3.1 starts with a length description of the physiology of subcortical auditory relay stations, which could be made more concise.

Could the authors provide their opinion on why it is important to parcellate the human auditory cortex in the first place (some neurophysiologists argue that this would not, necessarily, extend our knowledge on how auditory information is processed in the brain)?

More generally, it would also be nice to hear the authors' opinion and assessment on how likely it really is that entirely new neuroscience conclusions can be made based on laminar/columnar UHF fMRI, beyond replication of what is already known based on animal neurophysiology. What are the critical barriers for this progress?

Section 3.3 is highly interesting. Some slight refocusing might be needed on times, however, to link the major arguments to the topic of benefits achievable with UHF imaging.

There are a lot of acronyms, less would be better.

Author Response Letter

Reviewer 1

Very interesting paper. I don't have major comments apart from suggestions for increasing the focus, or minor editing remarks.

Response: We thank the reviewer for the suggestions and for the detailed editing corrections.

- *The review of the sparse sampling sequences is very interesting. I wonder whether it might help the reader to have some graphical representation of the advantage of sparse sampling vs. e.g., a noisy EPI sequence, but with a very fine temporal sampling (and more data points for a GLM-based estimation of BOLD responses). If noise is detrimental even when the fMRI time series is sampled very finely in time, then one should see clear differences in this comparison.*

Response: The effect of the various experimental setups on statistical results is an interesting topic. Previous work (with at TR = 2.2 s) showed that a sparse and clustered paradigm detected more sound activation than a continuous paradigm, and that the response amplitude to frequencies coinciding with scanner noise was in all acquisitions lower than that in the sparse paradigm (Langers et al., 2014).

It is not straightforward to predict the advantages/disadvantages of the various acquisition paradigm at a higher temporal resolution than the 2.2 s employed by Langers et al. (2014), as it requires, for example, taking into account the effect of serial autocorrelation. A comparison between acquisition paradigms can be done, but it requires collecting data. Instead, here we simply describe the different approaches, their advantages and disadvantages, and what considerations should be taken into account when designing study at 7T. We have now added the following sentence to the manuscript (p. 6):

“It would be of interest to experimentally compare the sensitivity of continuous, clustered, and sparse acquisition techniques combined with fast temporal sampling as made feasible by MB EPI.”

- *The review of UHF applications to the imaging of subcortical structure of the auditory system is remarkable and very well justified.*

Response: We thank the reviewer for this comment.

- *Part of the review of the category vs. acoustics studies is about limitations in variance partitioning methods and in model-matching methods. I do love these considerations, and I think they are important. However, I think they start decreasing the focus of the manuscript from the main topic: how does UHF help auditory neuroscience? A better justification for these considerations might be appropriate. Again, most of the section is about non UHF literature, and the majority of UHF-related considerations are in large part about future research directions.*

Response: We thank the reviewer for this useful comment. Following her/his suggestion, we have restructured section 3.3. We now first introduce the issue of assessing the relevance of acoustic processing in higher level auditory areas (such as in STG/STS), and state that UHF can help resolving this issue. In combination with adequate computational approaches UHF allows (page 15):

“approaching the correct scale at which these computations are performed”.

In particular, to make a case for higher resolution acquisition, we explain how the bias towards the cortical surface of fMRI responses collected at low field strength and/or low resolution may (page 15):

“overrepresent the output of intra-cortical computations or the effects of feedback to a cortical area.”

To highlight how these biases may have impacted our current understanding of auditory cortical processing, we next review the key features of opposing views on the function carried out by regions in the STG/STS. This requires discussing methodological aspects other than the magnetic field of acquisition or imaging resolution. We agree with the reviewer that the original discussion of these issues was too detailed. Following the reviewer’s comment, we have simplified the section summarizing the key features of recent computational studies investigating the processing in STS/STG. We have, for example, removed references to variance partitioning or other technical details (e.g., the use of Ridge regression or regularization). Nevertheless, it is important to stress how the differences in conclusions obtained from different approaches may not be entirely warranted. This allows us to moderate the current discussion based on analysis methods (page 16):

“A safer interpretation of the computational fMRI studies carried so far across field strengths is that low-level acoustics influence STG/STS responses (Santoro et al., 2014), but part of the response in these regions can be explained by a functional specialization that reaches beyond low-level acoustics (Norman-Haignere and McDermott, 2018).”

With this in mind, we propose that categorical representations are the result of intra-cortical processing within an area (across layers). This idea is drawn as a parallel to the known laminar processing of PAC that we have recently confirmed using UHF in humans (page 17):

“Due to the arrival of feedback signals as well as intra-cortical (across laminae; within a column) processing, the sound representation in superficial and deep layers may be more complex than in middle input layers (Hirsch and Martinez, 2006; Linden, 2003). In PAC the increase in processing complexity through cortical layers has been shown both using electrophysiology in cats (Atencio et al., 2009) as well as in humans with UHF laminar fMRI (Moerel et al., 2019). In STG/STS, middle layers may partially represent the acoustic content while superficial layers could show an

increasingly categorical response due to intra-cortical processing across layers (or feedback).”

We believe these changes highlight the relevance of UHF for the understanding of cortical auditory processing, by merging recent results and suggesting a way to resolve the debate on the relevance of acoustics in the lateral temporal cortex.

Detailed remarks:

- P3, L16: *[small subcortical structures] give example, and approximate volume estimate?*

Response: We have added volume estimates for the other subcortical auditory structures in section 3.1 (page 9):

“Size estimates of the subcortical auditory structures vary. Ex vivo estimates are considerably lower (CN = 46 mm³, SOC = 7 mm³, IC = 65 mm³, and MGB = 58 mm³; Glendenning and Masterton, 1998) than in vivo probabilistic estimates (e.g., CN = 24 mm³, SOC = 63 mm³, IC = 189 mm³, and MGB = 207 mm³ based on significant responses to sounds in at least 4/10 participants; Sitek et al., 2019).”

- P3, L16: *[inner workings] a bit of an approximate description. What about "that could not be resolved in detail at lower fields"?*

Response: We have changed the text following the reviewer’s suggestion.

- P3, L27: *[In addition... entire paragraph] the usefulness of these improvements in UHF anatomical imaging to auditory neuroscience are not crystal clear. Could you elaborate?*

Response: We have removed the paragraph, as the main focus of the review is on fMRI. This furthermore addresses a later comment of this reviewer, as removing this paragraph substantially shrinks the introduction.

- P4, L12: *[uniquely human traits (i.e., speech and music)] speech and music are not traits. In general, I find this entire sentence a bit unnecessary: non-invasive exploration of the auditory system is not limited to UHF imaging. And I am not sure you need to convince the readership of your paper that audition is important. They wouldn't open it if they weren't convinced already ;-)*

Response: Following the suggestion of the reviewer, we have removed this sentence and the next one (referred to in the next comment).

- P4, L12: *[driven by these unique demands... rest of sentence] speech is unique to humans but vocal communication isn't. In general, reducing the importance of hearing to speech and music is an overstatement. Otherwise we could cross a road while wearing a muffler and still survive, or walk on glasses that were just broken out of your sight (but not out of our hearing) and still not hurt ourselves... Again, I am not sure it is important to convince readers here that audition is important. Finally, if we consider hearing as object processing through a non-visual modality (that goes beyond corners like smell but no other), it becomes clear that important hearing abilities are shared throughout much of the animal kingdom.*

Response: Following the suggestion of the reviewer, we have removed this sentence.

- P4, L18: *[Here, we review the use of UHF MRI for studying the human auditory system] Here, we review auditory neuroscience studies based on UHF MRI.*

Response: We have changed the text following the reviewer's suggestion.

- P4, L21: *[First,... rest of paragraph] The introduction is slightly redundant, surely starting from here. I would personally think that making this points briefly right away in the first paragraph of the MS (along with some shrinking of the subsequent content, if possible) might further streamline the introduction.*

Response: We have changed the text following the reviewer's suggestion.

- P4, L22: *[enabling studying how sounds are processed] enabling the study of sound processing*

Response: We have changed the text following the reviewer's suggestion.

- P4, L25: *[may be localized individual hemispheres] I would try to be more specific here: someone naive to UHF imaging might think that it's already possible to localize cortical regions on each hemisphere, at least approximately, using e.g., anatomical atlases.*

Response: We have added the following sentences to clarify the current challenges in the localization of auditory cortical regions (page 3):

“Second, the versatility of MR contrasts allows assessing a range of anatomical and functional brain characteristics and may thereby enable the in vivo and individually-based localization of human auditory cortical regions. This is not trivial, as the vast interindividual variation in superior temporal lobe macroanatomy severely limits the localization accuracy of auditory regions through probabilistic atlases.”

- P4, L26: *[study the columnar computations] "shed light on column specific computations" would be more acceptable. We don't study computations directly, particularly with fMRI (regardless of the field strength).*

Response: We have changed the text following the reviewer's suggestion.

- P4, L27: *[This may show an abstract] not really something that can be done with UHF fMRI only.*

Response: This statement was meant in reference to the contribution of layer-dependent (within an area) computations in the emergence of a categorical representation. While role of M/EEG and ECoG is important for studying across-area computations (but see Bonaiuto et al., 2018), this is not what we refer to in this sentence. To clarify, we have changed the sentence as follows (page 3):

“Third, UHF fMRI may shed light on column specific computations (i.e. across layers) that take place throughout human auditory cortex. This may show how layer-dependent computations contribute to the emergence of an abstract, categorical sound representation from low-level acoustic feature processing.”

- P4, L29: *[understanding] mechanistic understanding*

Response: We have changed the text following the reviewer's suggestion.

- P5, L5: *[so do the gradients move] do gradients actually move? Or is something else whose movement is transformed into acoustical energy?*

Response: A current applied through the gradient coil in the presence of a magnetic field generates forces that act on the gradient coil. The end result of this is mechanical vibrations, and thus movement of the gradient coil/assembly, which in turn produces the acoustic noise (much like the effect of an audio speaker vibrating; Yao et al., 2004). The size of the effect depends on the elastic properties, weight, and length/size of the coil as well as magnetic field strength. Other objects in/near the bore (like the patient table, RF coil, etc.) could be impacted by acoustic/vibrational effects from the gradients, depending on their respective physical properties/resonances.

- P5, L10: *[in which the forces were compensated locally] some more specifics would be interesting given the importance of the issue.*

Response: While important, this also concerns a rather complicated engineering topic that is beyond the scope of the review. Following the reviewer's suggestion, we have elaborated in the text as follows (page 4):

"Resulting forces depend on the angle of the conductor relative to the magnetic field, as well as the current direction. The design of a gradient coil in which the forces were compensated locally (i.e., with multiple current generating loops that result in forces that cancel each other), was perhaps one of the first technological developments that benefitted auditory fMRI research. This gradient coil design reduced the overall vibration of the gradient coil assembly and thus the resulting acoustic noise (Mansfield et al., 1994)."

- P5, L12: *[attenuating the scanning noise... Sensimetrics] does Sensimetrics use active noise cancellation? If not, how is noise attenuation achieved?*

Response: The Sensimetric system does not use active noise cancellation. Compared to very early systems, which for example used ambient sound presentation, noise attenuation is achieved by combining the presentation of sounds in the ear canal with good earplugs and additional ear muffs. This allows for noise attenuation compared to actual sound delivery. This is now clarified by changing the text as follows (page 5):

"Most systems used nowadays in both low and high field imaging achieve noise attenuation, compared to actual sound delivery, by combining the presentation of sounds in the ear canal with high quality earplugs and ear muffs (see e.g. the Sensimetric system; www.sens.com)."

- P5, L23: *[which has] and has*

Response: We have changed the text following the reviewer's correction.

- P5, L23: *[advantage of fully separating the BOLD response to the sound from the BOLD response to the scanner noise] this would be true if e.g., one estimated the BOLD response to the scanner noise and the BOLD response to each sound stimulus with HRF-convolved regressors for each. Is this usually the case in HRF fMRI for auditory neurosciences? Also, you might specify "sound stimulus" instead of "sound" only.*

Response: In sparse sampling, the TR is in the order of 12-20 s. As a result, the BOLD response to the scanner noise is simply not measured (eliminating the need/possibility to

estimate it). This is shown in Figure 1C by the lack of overlap between the red dotted line (the BOLD response to the scanner noise) and the grey box (the MR acquisition). We have slightly amended the sentence to clarify this issue (page 5):

“...has the advantage of fully separating (in time) the BOLD response to the sound from the BOLD response to the scanner noise (the latter is not measured due to the very long TR).”

- P5, L26: *[severely limiting the power of the experimental design] or severely increasing the uncertainty of GLM-based (etc.) BOLD estimates because of the reduced number of data points?*

Response: The reviewer points out one of the reasons for the reduced power. We have added this to the text in order to make our argument more specific:

“..., severely increasing the uncertainty of GLM-based BOLD estimates due to the reduced number of data points and thereby limiting the power of the experimental design.”.

- P6, L15: *[on the order] in the order*

Response: We have changed the text following the reviewer’s correction.

- P6, L29: *[https:...] is this the right citation format for this resource?*

Response: We have replaced the citation by Uğurbil et al. (2013).

- P7, L10: *[necessitate] require*

Response: We have changed the text following the reviewer’s suggestion.

- P7, L12: *[specific absorption rate] a definition would help*

Response: We have changed the text as follows (page 8):

“... limited by the specific absorption rate (SAR; i.e., the radio frequency power absorbed per unit of mass of tissue).”

- P7, L15: *[et al. [2012] for] et al., 2012, for*

Response: We have changed the text following the reviewer’s suggestion.

- P7, L17: *[Furthermore] not a great way to start a new paragraph. An alternative such as "another issue" or similar would be better*

Response: We have changed the text following the reviewer’s suggestion.

- P11, L16: *[(communication)] why this particular class of sounds?*

Response: There is no reason to refer to this sound class in particular. We have removed the word ‘communication’ from the sentence.

- P13, L15: *[Alternatively,] similar remark as above for paragraph starting with "Furthermore". Perhaps remove "alternatively".*

Response: We have changed the text following the reviewer's suggestion.

- P16, L10: *[Second, the principles...] might be worth it to explicitly state that this regularization is required to do e.g., encoding analyses, whereas OLS variance partitioning is fine in the absence of regularization requirements.*

Response: The reviewer correctly points out that variance partitioning (this and the next point) is a challenge when regularization is used. In response to a more general comment on this section raised before, we have shortened and simplified this section. As a result, we do not refer anymore to the issues introduced by regularization. The main issue we want to raise is the one that it is difficult to assess the effects of acoustics based on responses to natural sounds (page 16):

"As a result, there may be a bias when assessing the relevance of acoustic processing when that is based on an acoustic model's prediction accuracy of responses to natural sounds."

While solutions for variance partitioning exist for regularized models, we believe that this simplified treatment of the topic aids its readability.

- P16, L14: *[their performance under crossvalidation] this is again specific to encoding. Cross-validated variance partitioning in the absence of regularization should be much more straightforward...*

Response: We thank the reviewer for the comment. As for the previous comment, this issue has been taken care of by shortening the section (see previous point).

- P18, L16: *[these include pitch, timbre and harmonics-to-noise-ratio] I believe other fMRI studies not mentioned here, and predating the majority of those in these list, investigated all these features at once, and some more as well.*

Response: We thank the reviewer for pointing out this omission. After restructuring the discussion in response to an earlier comment of this reviewer, it became unnecessary to discuss these more complex acoustic features. For this reason, this section has been removed from the manuscript.

Reviewer 2

Summary:

In this work, the authors review the state-of-the-art of ultra-high field MRI for studying the human auditory system non-invasively. The manuscript is well-written and organized, and represents a comprehensive overview from researchers who have been responsible for many key studies that now make the technique more accessible to other researchers. It will likely serve as an important reference for auditory researchers in the coming years.

My comments are relatively minor and mainly consist of suggestions for flow and consistency, see below.

Best regards,

Emily Coffey

Comments:

1. *Part of the purpose of this work seems to be to encourage a larger segment of the auditory neuroscience community to adopt 7T for a subset of research questions for which it is uniquely suited. While the explanation of the opportunities are very convincing, this 3T reader started to feel in need of some reassurance concerning the accessibility of the technique for wider application in auditory cognitive work, specifically around page 7. It could be very helpful for users to have something like a "best practices" "starting point recommendations" or decision tree diagram (maybe for a few common designs?), inset box or paragraph to provide some advice as to where to get started making experimental/scan parameter decisions. Failing that, perhaps at least a sentence or two to the effect that while many of these matters need further study, we are at the point where some design choices are known/semi-optimized such that cognitive work can proceed.*

Response: We thank the reviewer for the thoughtful comments. We certainly would like to stimulate auditory neuroscientists to transfer from 3T to 7T if beneficial to their research question. We have added Table 1 summarizing strengths and weaknesses of the main functional imaging sequences (page 38) as well as the following section to get them started (page 8):

“Compared to early work (Formisano et al., 2003), the transition from 3T to 7T has become much easier. Tested imaging sequences and protocols for both functional and anatomical imaging are readily available. The choice of the acquisition technique (sparse, clustered, continuous, or ISSS) follows the same considerations as those followed at 3T, amounting to the trade-off between the biasing influence of scanner noise compared to the number of collected data points. The choice of imaging sequences for studying the auditory cortex is dominated by trade-off in SNR, coverage, temporal resolution, and spatial specificity, as detailed in Table 1. Imaging of subcortical auditory structures necessitates obtaining images with high spatial resolution. Combined with the low SNR in central brain regions, this currently limits the choice to GE-EPI acquisitions for subcortical auditory imaging.”

2. *I suggest to include a brief overview of the organization of the rest of the paper at the end of the introduction to orient readers.*

Response: Following the reviewer’s suggestion, we have added the following section to the end of the introduction (page 4):

“Below we first review methodological challenges and then discuss the potential benefit of UHF MRI for investigating subcortical auditory processing, parcellating the human auditory cortex non-invasively, and studying the emergence of categorical responses in the auditory cortex.”

3. *Pg. 8, concluding paragraph to section 2 was about solutions for scanner noise - I think this would make sense placed after the paragraph about scanner noise being the main challenge for auditory UHF. The point about HF driving technology didn't come across very clearly for me and I don't think is a critical one, so could be removed. The sentence that starts with "Despite the improvements (...)" also seemed to be setting up for an argument that didn't really happen, could be reworded to just emphasize that things are pretty good but will get better, and here are some of the ways.*

Response: Thank you for the suggestion. We have modified the text accordingly.

4. Pg. 9, subcortical structure overview: *"The partitions of the CNs (...)"* - this long sentence is hard to parse as the structure of the phrases is not parallel, suggest to highlight function and structure in the same order and form for each. The sound localization / SO point is also introduced twice with an intervening point about CN, perhaps just reorganize them according to ascending order of structures. IC is introduced as an "obligatory relay station". The authors clearly don't mean that it only does that, and specifically make the point for the MGB that it's not only a relay just a few lines after, but as some of the auditory subcortical people have been fighting against the idea that the IC is just a relay, it could be better to use another word. "regulates information flow" is intriguing but vague, perhaps an example or two of what that means from those papers?

Response: We thank the reviewer for pointing this out, and have changed the text accordingly (page 9):

"The CN consists of partitions that differ in cell morphology, response characteristics, and connectivity. These partitions have been differentially implicated in the detection of timing cues (Tollin, 2003), cues relevant to sound localization in the vertical plane (dorsal CN - Oertel and Young, 2004; Reiss and Young, 2005), and the extraction of pitch and intensity information from complex sounds (Blackburn and Sachs, 1990; Kim et al., 1986)."

And page 10:

"The inferior colliculus regulates information flow ascending from the brainstem nuclei to the thalamus and cortex."

5. Pg. 13 *"To date, none of these functionally-based options (...)* has been adopted by the community" - why not? As an auditory researcher I sometimes struggle to decide which of the many parcellations to use, so I am intrigued to know why the authors think the poor adoption is the case, and it would better help me understand why UHF could help add to the discussion, which is the subject of the subsequent paragraph. Do they authors think this is because they are not seen to be reproducible, vary too much by analysis technique, too hard to implement, or other factors?

Response: This is a very interesting question that we have often discussed about amongst ourselves. We believe that it results from a combination of factors. First, none of the available parcellation schemes that are based on functional mapping have been validated against a "ground truth" coming from microanatomy (e.g. cytoarchitecture). This endeavor is complicated by the large variability in temporal lobe macroanatomy and the relatively poor understanding of how this macroanatomical variability impacts microanatomy. Finally, at the microanatomical level there are several parcellation schemes that do not agree with each other. These issues are discussed in section 3.2 together with indications of how UHF can help. In summary, our opinion is that ultra-high field MRI can help by providing in vivo information not accessible at 3T. This includes anatomical information (e.g., layer-dependent myelin-related cortical contrast, radially information) and layer-dependent differences in functional computations. This information, if complemented with methods that can account for large macroanatomical differences, can be leveraged to obtain a more accurate definition of cortical areas that will ultimately have to meet the test of validation against microanatomical definitions.

6. Pg. 14 the section about UHF options that differ in sensitivity seemed more similar to the challenges/methods material in section 2, as it is quite methodsy and might distract from the focus of section 3. It could be moved to section 2, perhaps renaming that "Auditory specific methodological considerations" (to include both challenges and positive points),

and then the text in 3 could focus more on the neuroscience questions and quickly refer to the fact that there are scanning sequences that can get at those questions. However, this is only a suggestion that would appeal to me organizationally and I leave it to the authors.

Response: We have reorganized the text following the reviewer's suggestion.

7. *P.11 + The authors have some excellent suggestions for next steps, but they are scattered around the end of the manuscript. If the journal format allows, it could be nice to also collect them into a point-form Box, similar to how TiCS papers are organized.*

Response: Following the reviewer's suggestion, we have added a Box with suggested next steps to the manuscript (page 39).

8. *P. 20 The final paragraph (about clinical advances) seems a bit tacked on there and is not related directly to the title and main focus of the paper. Though it is nice to include, I would suggest to poke that material in somewhere else, maybe even in the bottom of the introduction and then say that while clinical advances are promising and increasing, you're going to focus on understanding representation (i.e. basic science) in the rest of the paper. The last sentence in the penultimate paragraph is a nice strong positive conclusion about UHF and auditory neuroscience.*

Response: We have reorganized the text following the reviewer's suggestion.

Very minor:

- *Use of the word "Interestingly," seems slightly odd contextually in two places, on pg. 6 and 7 - nothing is lost by its removal*

Response: Thank you. We removed the word 'Interestingly'.

- *Check consistency of "in vivo" vs. "in-vivo", and I believe depth-dependent should be hyphenated, though it may be a stylistic choice*

Response: Thank you for noticing. We made sure to stay with "in vivo" and "depth-dependent" throughout the text.

- *Pg. 6 "An exception to this is represented" - due to length of preceding sentence, this reader had to check what the referent was, could make clearer*

Response: We have rephrased the sentence as (page 6):

"An exception to this is the required use of continuous acquisition when presenting long sound excerpts, such as audio books (Hanke et al., 2014) or movies (Uğurbil et al., 2013)."

- *Pg. 9. "As the IC," - word missing? "As in the IC," or "Analogous to the IC"?*

Response: We meant to say "Just like the IC,...", but have removed that part of the sentence to increase readability.

- *Pg. 18. "This may resolve conflicting results across previous studies" - (e.g. REFs) would be helpful here*

Response: We have added references to the paper of Norman-Haignere and McDermott (2018) and Santoro et al. (2014).

Reviewer 3

This is a high-quality review of the potential applications of ultra-high field (UHF) anatomical and functional MRI in studies of human auditory system, a domain that could benefit from UHF imaging even more than research on the much more intensively studied visual system. The authors offer interesting perspectives on how to deal with obstacles of using UHF fMRI in studies of the auditory system, such as the more general problem of acoustical noise, as well as the more specific challenges such as deficiencies in transmit B1 profiles in or near the superior temporal plane. On the other hand, the manuscript communicated many new ideas and guidance on research areas where future UHF studies could be most fruitful, such as the functional mapping of subcortical auditory pathways and the functional parcellation of human auditory cortices. This review will likely be very well received by the human auditory neuroimaging community. I do not have any major concerns or objections, but I have listed a few more specific suggestions below.

Specific comments:

The manuscript could be made even more interesting and useful for those who are just entering the field of UHF MRI of auditory system by making it more assertive or opinionated. For example, the authors could more clearly express their assessment of available scanning approaches, including conventional continuous, sparse sampling, clustered sampling, and ISSS, as well as of the potential trade-offs using approaches such as SWIFT. How about smoothing?

Response: We thank the reviewer for this comment, which is very much in line with the comment of reviewer 2. We believe that choices for both the way sounds are presented (sparse sampling, continuous etc.) or acquisition sequences (GE-EPI, VASO) is not about availability, but about trade-offs. For the sound presentation schemes, these trade-offs are no different at 7T than 3T. The main differences have been reported elsewhere, and we now refer to them in a sentence (page 5):

“In a direct comparison between acquisition techniques, it was shown that the response amplitude to frequencies coinciding with scanner noise was underestimated by continuous and clustered acquisition, but not when using a sparse design (Langers et al., 2014).”

Some quantitative comparisons are missing, and we now indicate the potential interest in performing them (page 6):

“It would be of interest to experimentally compare the sensitivity of continuous, clustered, and sparse acquisition techniques combined with fast temporal sampling as made feasible by MB EPI.”

Regarding the choice of the acquisition, if one is willing to sacrifice coverage and temporal resolution, it is possible to gain in spatial specificity. We have now provided an overview of the advantages and disadvantages of the available options in Table 1 (page 38). Our opinion is summarized at the end of section 2 (page 8) in order to hopefully support auditory neuroscientist entering the field of ultra-high field MRI.

Regarding the use of SWIFT, we believe this is a very interesting opportunity. However, it will require development to assess, e.g., its functional specificity and sensitivity. We have included this in Box 1 that summarizes what we believe are the next steps for auditory neuroscience at UHF (page 39).

Finally, we have decided not to focus on considerations regarding the analyses of the data in this review. These considerations are not specific to auditory neuroscience and have been reviewed elsewhere (see, e.g., Polimeni et al., 2018).

The section 3.1 starts with a length description of the physiology of subcortical auditory relay stations, which could be made more concise.

Response: Considering the anatomical and functional complexity of these regions, section 3.1 is already quite short. We would prefer not to shorten it too much, as understanding the challenge (e.g., knowing that each of these small regions consists of functionally distinct subdivisions) is important in order to understand the benefit of ultra-high field (f)MRI. However, in response to the reviewers' comments, we have tried to improve the readability of the section.

Could the authors provide their opinion on why it is important to parcellate the human auditory cortex in the first place (some neurophysiologists argue that this would not, necessarily, extend our knowledge on how auditory information is processed in the brain)?

Response: We thank the reviewer for the interesting question. We have added the following argument to the manuscript (Box 2; page 39):

“Box 2. What is the neuroscientific value of parcellating the human auditory cortex?

Efforts to parcellate the human auditory cortex into distinct subfields based on in vivo measures are abundant (Barton et al., 2012; De Martino et al., 2015b; Dick et al., 2012; Moerel et al., 2012), yet the value of obtaining such a parcellation for auditory neuroscience may be questioned. That is, one may argue that a parcellation would not, necessarily, extend our knowledge on how auditory information is processed in the brain. Here we argue in favor of the value of obtaining a parcellation. Our rationale follows from the hypothesis that an accurate cortical parcellation identifies areas with unique functional properties. The ability to identify such areas across individuals (and studies) would allow merging results in a common reference space. Such an approach would substantially reduce the variability introduced by, e.g., the variable macroanatomy of the human temporal lobe. Without a common reference space, or using a reference space that does not correctly align functionally identical regions to each other (e.g., through the use of a probabilistic atlas), neuroscientific effects could be blurred or even removed. This is especially true for UHF MRI applications in which, due to the push for high spatial resolution, signal-to-noise ratio (SNR) is limited. In cortical depth dependent studies, for example, averaging (or spatially smoothing) data within functionally coherent areas (while maintaining separate the signal across depths) is a common practice used to increase SNR. This requires the area to be defined correctly defined to avoid averaging across regions with functionally distinct properties. The areal definition also needs to be performed in an unbiased manner in order to avoid double-dipping. While functional localizers (or the combination of functional and anatomical data) can be used to define some auditory regions (e.g., voice regions; (Belin et al., 2000), such localizers are not available for the majority of the auditory cortex. Obtaining a parcellation of human auditory cortex would allow the unbiased definition of areas. Furthermore, it would allow the assignment of homologies across species and thereby bridge results obtained using non-invasive measurements to the human brain.

More generally, it would also be nice to hear the authors' opinion and assessment on how likely it really is that entirely new neuroscience conclusions can be made based on laminar/columnar UHF fMRI, beyond replication of what is already known based on animal neurophysiology. What are the critical barriers for this progress?

Response: Following the reviewer suggestion we have added the following argument to the manuscript (Box 3; page 40):

“Box 3. Can laminar and columnar ultra-high field fMRI studies contribute neuroscientific knowledge?”

Is it possible to draw entirely new neuroscience conclusions from laminar and columnar UHF fMRI studies, beyond the replication of what is already known from animal neurophysiology? This critical question is perhaps not surprising, as most of the early UHF fMRI work has focused on replicating known neuroscientific organizations (e.g., ocular dominance and orientation columns in primary visual cortex (Yacoub et al., 2007); tonotopic columns in primary auditory cortex (De Martino et al., 2015a)). Even to date, known neuroscientific organizations are studied in order to validate the employed methodology or evaluate strengths and weakness of novel UHF techniques (De Martino et al., 2013; Huber et al., 2018; Kemper et al., 2015; Moerel et al., 2018). However, recent years have also seen applications beyond early cortical regions shedding light on the intra-cortical computations relevant to complex tasks (e.g. working memory; Finn et al., 2019). The main strength of ultra-high field fMRI lies in the combination of the spatial resolution required to access the mesoscopic scale (accessing feedforward and feedback processing through subdivisions of subcortical regions and cortical layers), while maintaining the coverage required to image a large part of brain. Thereby, this tool is optimally suited to gain an understanding of intra-areal interactions and processes. The ability to measure high spatial resolution signals with large coverage enables studying how information is transformed between brain regions. In auditory neuroscience, imaging the complete auditory pathway at a high spatial resolution allows investigating how sound representations are modified by task, learning, and disease. UHF fMRI is unique in this respect, and its results will surely provide novel neuroscientific insights. In particular, we expect the testing of theories that make specific assumptions regarding the mesoscopic implementation level of brain processes and/or rest on the hypothesis of hierarchical interactions across regions (e.g. predictive coding) in the near future.”

Section 3.3 is highly interesting. Some slight refocusing might be needed on times, however, to link the major arguments to the topic of benefits achievable with UHF imaging.

Response: Following the suggestion of this reviewer, as well as that of Reviewer 1, we have restructured the section to highlight the role of 7T in resolving this debate.

There are a lot of acronyms, less would be better.

Response: We agree, but unfortunately the field of MRI is full of acronyms. We have already tried to limit their use. We feel that, for those familiar with the acronyms, removing more acronyms may make the text more confusing/ less readable.

References:

- Atencio, C.A., Sharpee, T., Schreiner, C.E., 2009. Hierarchical computation in the canonical auditory cortical circuit. *Proc. Natl. Acad. Sci.* 106, 21894–21899. <https://doi.org/10.1073/pnas.0908383106>
- Barton, B., Venezia, J.H., Saberi, K., Hickok, G., Brewer, A.A., 2012. Orthogonal acoustic dimensions define auditory field maps in human cortex. *Proc. Natl. Acad. Sci.* 109, 20738–20743. <https://doi.org/10.1073/pnas.1213381109>

- Belin, P., Zatorre, R.J., Lafaille, P., Ahad, P., Pike, B., 2000. Voice-selective areas in human auditory cortex. *Nature* 403, 309–312. <https://doi.org/10.1038/35002078>
- Blackburn, C.C., Sachs, M.B., 1990. The representations of the steady-state vowel sound /ε/ in the discharge patterns of cat anteroventral cochlear nucleus neurons. *J. Neurophysiol.* 63, 1191–1212. <https://doi.org/10.1152/jn.1990.63.5.1191>
- Bonaiuto, J.J., Meyer, S.S., Little, S., Rossiter, H., Callaghan, M.F., Dick, F., Barnes, G.R., Bestmann, S., 2018. Lamina-specific cortical dynamics in human visual and sensorimotor cortices. *Elife* 7. <https://doi.org/10.7554/eLife.33977>
- De Martino, F., Moerel, M., Ugurbil, K., Goebel, R., Yacoub, E., Formisano, E., 2015a. Frequency preference and attention effects across cortical depths in the human primary auditory cortex. *Proc. Natl. Acad. Sci.* 201507552. <https://doi.org/10.1073/pnas.1507552112>
- De Martino, F., Moerel, M., Xu, J., van de Moortele, P.-F., Ugurbil, K., Goebel, R., Yacoub, E., Formisano, E., 2015b. High-Resolution Mapping of Myeloarchitecture In Vivo: Localization of Auditory Areas in the Human Brain. *Cereb. Cortex* 25, 3394–3405. <https://doi.org/10.1093/cercor/bhu150>
- De Martino, F., Zimmermann, J., Muckli, L., Ugurbil, K., Yacoub, E., Goebel, R., 2013. Cortical Depth Dependent Functional Responses in Humans at 7T: Improved Specificity with 3D GRASE. *PLoS One* 8, e60514. <https://doi.org/10.1371/journal.pone.0060514>
- Dick, F., Tierney, a T., Lutti, a, Josephs, O., Sereno, M.I., Weiskopf, N., 2012. In vivo functional and myeloarchitectonic mapping of human primary auditory areas. *J Neurosci* 32, 16095–16105. <https://doi.org/10.1523/JNEUROSCI.1712-12.2012>
- Finn, E.S., Huber, L., Jangraw, D.C., Molfese, P.J., Bandettini, P.A., 2019. Layer-dependent activity in human prefrontal cortex during working memory. *Nat. Neurosci.* 22, 1687–1695. <https://doi.org/10.1038/s41593-019-0487-z>
- Formisano, E., Kim, D.-S., Di Salle, F., van De Moortele, P.-F., Ugurbil, K., Goebel, R., 2003. Mirror-Symmetric Tonotopic Maps in Human Primary Auditory Cortex. *Neuron* 40, 859–869. [https://doi.org/10.1016/S0896-6273\(03\)00669-X](https://doi.org/10.1016/S0896-6273(03)00669-X)
- Glendenning, K.K., Masterton, R.B., 1998. Comparative Morphometry of Mammalian Central Auditory Systems: Variation in Nuclei and Form of the Ascending System. *Brain. Behav. Evol.* 51, 59–89. <https://doi.org/10.1159/000006530>
- Hanke, M., Baumgartner, F.J., Ibe, P., Kaule, F.R., Pollmann, S., Speck, O., Zinke, W., Stadler, J., 2014. A high-resolution 7-Tesla fMRI dataset from complex natural stimulation with an audio movie. *Sci. Data* 1. <https://doi.org/10.1038/sdata.2014.3>
- Hirsch, J.A., Martinez, L.M., 2006. Laminar processing in the visual cortical column. *Curr. Opin. Neurobiol.* <https://doi.org/10.1016/j.conb.2006.06.014>
- Huber, L., Tse, D.H.Y., Wiggins, C.J., Uludağ, K., Kashyap, S., Jangraw, D.C., Bandettini, P.A., Poser, B.A., Ivanov, D., 2018. Ultra-high resolution blood volume fMRI and BOLD fMRI in humans at 9.4 T: Capabilities and challenges. *Neuroimage* 178, 769–779. <https://doi.org/10.1016/j.neuroimage.2018.06.025>
- Kemper, V.G., De Martino, F., Vu, A.T., Poser, B.A., Feinberg, D.A., Goebel, R., Yacoub, E., 2015. Sub-millimeter T2 weighted fMRI at 7 T: comparison of 3D-GRASE and 2D SE-EPI. *Front. Neurosci.* 9, 163. <https://doi.org/10.3389/fnins.2015.00163>
- Kim, D.O., Rhode, W.S., Greenberg, S.R., 1986. Responses of Cochlear Nucleus Neurons to Speech Signals: Neural Encoding of Pitch, Intensity and other Parameters, in: *Auditory Frequency Selectivity*. pp. 281–288. https://doi.org/10.1007/978-1-4613-2247-4_31
- Langers, D.R.M., Sanchez-Panchuelo, R.M., Francis, S.T., Krumbholz, K., Hall, D.A., 2014. Neuroimaging paradigms for tonotopic mapping (II): The influence of acquisition protocol. *Neuroimage* 100, 663–675. <https://doi.org/10.1016/j.neuroimage.2014.07.042>

- Linden, J.F., 2003. Columnar Transformations in Auditory Cortex? A Comparison to Visual and Somatosensory Cortices. *Cereb. Cortex* 13, 83–89. <https://doi.org/10.1093/cercor/13.1.83>
- Moerel, M., De Martino, F., Formisano, E., 2012. Processing of Natural Sounds in Human Auditory Cortex: Tonotopy, Spectral Tuning, and Relation to Voice Sensitivity. *J. Neurosci.* 32, 14205–14216. <https://doi.org/10.1523/JNEUROSCI.1388-12.2012>
- Moerel, M., De Martino, F., Kemper, V.G., Schmitter, S., Vu, A.T., Uğurbil, K., Formisano, E., Yacoub, E., 2018. Sensitivity and specificity considerations for fMRI encoding, decoding, and mapping of auditory cortex at ultra-high field. *Neuroimage* 164, 18–31. <https://doi.org/10.1016/j.neuroimage.2017.03.063>
- Moerel, M., De Martino, F., Uğurbil, K., Yacoub, E., Formisano, E., 2019. Processing complexity increases in superficial layers of human primary auditory cortex. *Sci. Rep.* 9. <https://doi.org/10.1038/s41598-019-41965-w>
- Norman-Haignere, S. V., McDermott, J.H., 2018. Neural responses to natural and model-matched stimuli reveal distinct computations in primary and nonprimary auditory cortex. *PLoS Biol.* 16. <https://doi.org/10.1371/journal.pbio.2005127>
- Oertel, D., Young, E.D., 2004. What's a cerebellar circuit doing in the auditory system? *Trends Neurosci.* <https://doi.org/10.1016/j.tins.2003.12.001>
- Polimeni, J.R., Renvall, V., Zaretskaya, N., Fischl, B., 2018. Analysis strategies for high-resolution UHF-fMRI data. *Neuroimage* 168, 296–320. <https://doi.org/10.1016/j.neuroimage.2017.04.053>
- Reiss, L.A.J., Young, E.D., 2005. Spectral edge sensitivity in neural circuits of the dorsal cochlear nucleus. *J. Neurosci.* 25, 3680–3691. <https://doi.org/10.1523/JNEUROSCI.4963-04.2005>
- Santoro, R., Moerel, M., De Martino, F., Goebel, R., Uğurbil, K., Yacoub, E., Formisano, E., 2014. Encoding of Natural Sounds at Multiple Spectral and Temporal Resolutions in the Human Auditory Cortex. *PLoS Comput Biol* 10, e1003412. <https://doi.org/10.1371/journal.pcbi.1003412>
- Sitek, K.R., Gulban, O.F., Calabrese, E., Johnson, G.A., Lage-Castellanos, A., Moerel, M., Ghosh, S.S., de Martino, F., 2019. Mapping the human subcortical auditory system using histology, post mortem MRI and in vivo MRI at 7T. *Elife.* <https://doi.org/10.7554/elife.48932>
- Tollin, D.J., 2003. The lateral superior olive: A functional role in sound source localization. *Neuroscientist.* <https://doi.org/10.1177/1073858403252228>
- Uğurbil, K., Xu, J., Auerbach, E.J., Moeller, S., Vu, A.T., Duarte-Carvajalino, J.M., Lenglet, C., Wu, X., Schmitter, S., Van de Moortele, P.F., Strupp, J., Sapiro, G., De Martino, F., Wang, D., Harel, N., Garwood, M., Chen, L., Feinberg, D.A., Smith, S.M., Miller, K.L., Sotiropoulos, S.N., Jbabdi, S., Andersson, J.L.R., Behrens, T.E.J., Glasser, M.F., Van Essen, D.C., Yacoub, E., 2013. Pushing spatial and temporal resolution for functional and diffusion MRI in the Human Connectome Project. *Neuroimage* 80, 80–104. <https://doi.org/10.1016/j.neuroimage.2013.05.012>
- Yacoub, E., Shmuel, A., Logothetis, N., Uğurbil, K., 2007. Robust detection of ocular dominance columns in humans using Hahn Spin Echo BOLD functional MRI at 7 Tesla. *Neuroimage* 37, 1161–1177. <https://doi.org/10.1016/j.neuroimage.2007.05.020>
- Yao, G.Z., Mechefske, C.K., Rutt, B.K., 2004. Characterization of vibration and acoustic noise in a gradient-coil insert. *Magn. Reson. Mater. Physics, Biol. Med.* 17, 12–27. <https://doi.org/10.1007/s10334-004-0041-0>